# An overview of the ocean data ecosystem

Maya Bloch Haimson[1, 2], Yoav Lehahn[1], Tomer Sagi[3]

[1.] Department of Marine Geosciences, Leon H. Charney School of Marine Sciences, University of Haifa, Haifa 31905, Israel.

[2.] Israel Oceanographic & Limnological Research Ltd (PCB). Tel-Shikmona, P.O.B. 2336, Haifa 3109701.

[3.] Department of Computer Science, Aalborg University. Fredrik Bajers Vej 7K, 9220 Aalborg East, Denmark.

*Correspondence* to: Yoav Lehahn (ylehahn@univ.haifa.ac.il)

**Abstract**

The oceans, covering approximately 70% of Earth's surface, play a pivotal role in climate regulation, biodiversity, and biogeochemical processes. The large and growing volume and complexity of ocean data, spanning diverse disciplines and formats, and dispersed across a wide range of sources, presents opportunities and challenges for advancing scientific research, informing policy, and addressing societal needs.

In this review paper we aim to create an easy-to-navigate map of the field of ocean data, enabling the reader to establish a broad understanding of the ocean data sector, and bridging gaps between different disciplines and levels of familiarity with ocean data. This is done through the concept of the "data ecosystem", which is used to describe the actors, organizations, and infrastructures involved in all aspects of the data value chain. We propose a structured ocean data ecosystem model as a method for comprehensive mapping of the ocean data market landscape. The proposed model consists of five key elements: stakeholders, societal elements, interoperability tools (such as standards and best practices), data sources and product offering, and emerging solutions. We provide an up-to-date analysis of ocean data sources and emerging solutions and a summary of relevant data standardization efforts such as marine standards, vocabularies, and ontologies. All this will promote the development of needs-based solutions, components, products, services, and technologies, thus contributing to the evolution of the ocean data ecosystem and promoting data-based ocean research.

**Keywords**

Data ecosystem, ocean data, marine data, Oceanography, data innovation, multidisciplinary, international

## 1. Introduction

Covering approximately 70% of Earth's surface, the oceans hold critical insights into Earth's climate, biodiversity, and biogeochemical cycling, making ocean research essential for advancing scientific knowledge (Soranno et al., 2015; Tanhua et al., 2019; Vance et al., 2019; UN Ocean Ocean Decade, Data & Information Strategy, May 2023), informing policies (Curry et al., 2021; Styrin et al., 2017) and improving the ability to provide human society with food and energy (Lehahn et al., 2016). The study of the ocean relies on a very large, and growing, number of multidisciplinary measurements, which are constantly performed worldwide using various crewed and autonomous platforms. The data resulting from this remarkable collection endeavor - generically termed *ocean data* - are highly voluminous (Soranno et al., 2015; Tanhua et al., 2019b), diverse (Durden et al., 2017; Tanhua et al., 2019b), and dispersed (Tanhua et al., 2019b).

Ocean data utilization have several key challenges, which can be roughly divided into three categories: 1. *Data handling*, 2. *Disparate data management structures,* and 3. *Data integration / Interoperability*. Data handling refers to challenges of storage, transfer, processing, and infrastructure development (Durden et al., 2017). This includes the difficulty in finding, accessing and processing existing data (Ramalli and Pernici, 2023; Tenopir et al., 2015; Tzachor et al., 2023), as well as the limitations of the "portal-download" model, where users actively search for and download data, requiring ocean scientists to be very familiar with data sources, formats, and processes (Buck et al, 2019). The challenge of d*isparate data management structures* refers to the fact that the vast amount and variety of data types and data sources are stored and managed within many separate and different types of data management infrastructures ((Brett et al., 2020; Tanhua et al., 2019b). Within these infrastructures, there is often an existing gap between the scientists producing the data and the end-users consuming the data (Tanhua et al., 2021), which causes gaps in answering user needs for data usage and their ability to generate insights. The challenge of *data integration / Interoperability* refers to the fact that the dispersed data management systems, as well as interoperability tools such as metadata, development of ontologies, common protocols, and best practices, are not yet sufficiently well developed and standardized (Brett et al., 2020; Sagi et al., 2020; Tanhua et al., 2019b), this poses challenges for data integration, which limits the uptake of data. In addition, the ability to scale data integration is limited by manual processes which require input from experts in diverse fields (Durden et al., 2017; Sagi et al., 2020; Soranno et al., 2015; Tanhua et al., 2019b). Other challenges include assuring data quality (Brett et al., 2020) and the willingness to share data (Brett et al., 2020; Lima et al., 2022).

Having clear characteristics of "big data" (Curry et al., 2021), the *ocean data* lifecycle consists of several stages, from collection and processing, through storage and sharing, to analysis and interpretation (Buck et al., 2019; Curry et al., 2021; Durden et al., 2017; UN Ocean Ocean Decade, Data & Information Strategy, May 2023). In its essence, ocean research is mainly focused on the two endpoints of the ocean data lifecycle, namely data collection and data analysis. With the amount of available data rapidly increasing and with data-driven approaches becoming more and more common, the ability to address key oceanographic questions increasingly depends on the ability to explore and gain insight from the vast amount of available ocean data. Consequently, present-day ocean research leans towards expanding beyond the basic tasks of data

collection and analysis, directly and indirectly addressing different aspects of the ocean data life cycle. A major challenge facing ocean scientists in this endeavor is to become familiar with the various actors involved in the different aspects of ocean data and with the complex ways by which they interact. A useful framework for addressing this challenge is through the concept of a *data ecosystem (Curry et al., 2021; Oliveira et al., 2019)*, which has been used for characterizing and mapping complex big data value ecosystems in various domains, such as the European big data value ecosystem (Curry et al., 2021), and open health data (Heijlen and Crompvoets, 2021).

To this end, this paper presents an overview of the ocean-data ecosystem and proposes a model that maps its key actors and their roles, with particular attention to data platforms and interoperability. The discussion is framed from the perspective of the marine researcher as end user, with the overarching goal of  facilitating synergetic work between the actors involved in different aspects of ocean data, thus improving the ability to utilize the vast amount of available ocean data to address important scientific questions.

The paper is organized as follows. The general notion of a *data ecosystem* is explained and discussed in the next section. We then map and model the ocean data ecosystem, describing its main actors (Sec. 3) and their roles (Sec. 4). In Sec. 5, we discuss our proposed ocean data ecosystem model, focusing on several key concepts that emerge from its analysis. The paper is concluded in Sec. 6.

## 2. Data ecosystem

### 2.1 Data ecosystem general definitions and examples

The notion of a data ecosystem has been discussed in a number of papers e.g. (Curry et al., 2021; Gelhaar et al., 2021; Harrison et al., 2012; Heinz et al., 2022; Oliveira et al., 2019; Styrin et al., 2017; ul Hassan and Curry, 2021). By and large, the "ecosystem" metaphor is used to characterize and explore the interdependency between the actors, organizations, and infrastructures involved in all aspects of the data value chain, from data collection to analysis and value generation (Harrison et al., 2012; Styrin et al., 2017).

As proposed by Oliveira et al. (2019), the data ecosystem consists of four main concepts: actors and roles, relationships and resources. The actors, or stakeholders, are entities such as enterprises, institutions, and individuals with specific roles in the ecosystem that consume, produce, or provide data and other related resources (e.g., software, services, and infrastructure). Each actor within the ecosystem can have one or more roles. Typical roles within a data ecosystem include data providers (e.g. data aggregators, integrators, developers, harmonizers, publishers, and storers), policymakers, standardization and regulation parties, data users (which usually represent the end-users of the data ecosystem), data intermediaries or service providers and others. Finally, each actor is connected to other actors through relationships.

Curry et al., (2021) have created the "periodic table of elements of big data value" as a classification system for the European big data value ecosystem, with the aim of increasing the competitiveness of European industries. Taking an

interdisciplinary approach, the authors identify four fundamental elements of the big data value ecosystem, namely: (1) ecosystem (e.g. stakeholders, roadmap, impact), (2) research and innovation (e.g. centers of excellence), (3) business, policy and societal (e.g. regulation, data-driven innovations, and business models) and (4) emerging (e.g. AI and data spaces). The European data ecosystem was further analyzed by Hassan and Curry (2021), who performed a data ecosystem stakeholder analysis, identifying the needs and drivers of stakeholders concerning big data in Europe.

Data ecosystems facilitate collaboration by enabling stakeholders to share data and services, enhancing research outcomes by extracting value from the increasing volume of shared data (Ramalli and Pernici, 2023). Domain-specific requirements may create a challenging setting for adopting data ecosystems. Ramali and Pernici (2023) describe the challenges of data ecosystems for scientific data and data management solutions, as well as the different structures of data ecosystem architectures, such as *centralized, federated,* or *distributed* data ecosystems. In a *centralized* data ecosystem, a single data management entity has control over the data. In a *distributed* system, the data are managed centrally, but stored and processed in a distributed manner, and the network may encompass multiple geographical locations. In a *federated* network, data management processes are also distributed, facilitating multiple and possibly geographically distributed networks to work together (Ramalli and Pernici, 2023).

For the case of open health data, Heijlen and Crompvoets (2021) have mapped a data ecosystem that consists of the following elements: Stakeholders and their interests (i.e. actors that use the data to create added value and eventually the consumers of products and services) information policies, data preparation activities (e.g. data quality assessment, metadata, and data formats), infrastructural elements (e.g. dataset access portals, data analysis tools or visualization tools) and drivers (e.g. global trends, stakeholders needs and data sharing requirements). Their model also includes dynamical interactions between the elements.

## 2.2 The concept of a data ecosystem in ocean research

The concept of an ocean data ecosystem has been addressed from different perspectives. The UN Ocean Decade ("Ocean Decade – The Science We Need For The Ocean We Want," n.d.; Ryabinin et al., 2019) defines the vision of an interconnected ocean data and information ecosystem as a globally distributed enabling environment that includes the frameworks, infrastructure, tools, capacity, and resources, thus involving interactions between technology and human communities. The UN Ocean Decade Data & Information Strategy (Intergovernmental Oceanographic Commission, 2023) identifies key components of this ocean digital ecosystem, including observations and data collection, end-user applications, analytics modelling and prediction, and data management and sharing.

A comprehensive overview of the ocean data sector has been provided by Tanhua et al. (2019a, 2019b, 2021), who reviewed recent developments in the technical capacity and requirement setting for a data management system in the frame of the Global Ocean Observing System (GOOS). These papers emphasize the importance of well-managed data management

systems for ensuring the data collected by the ocean observing systems are accessible for current and future uses. The
Framework for Ocean Observing (FOO) (deYoung et al., 2019; Lindstrom et al., 2012; Tanhua et al., 2019a), serves as a guideline for developing a multidisciplinary, integrated ocean observing system for operational purposes. It also refers to successes and challenges in its implementation and consider ways to ensure broader use of the Essential Ocean Variables (EOVs), providing a description of many of the actors of the ocean data sector. This includes SeaDataNet, EMODNET ("European Marine Observation and Data Network (EMODnet)," n.d.; Míguez et al., 2019), Environmental Research
Division's Data Access Program (ERDDAP) and the Argo program), as well as standards and semantic interoperability tools. (Tanhua et al., 2019a). They characterize the challenges of ocean data (wide diversity, disparate data sources, increased volume, and poorly defined best practices) and encouraged the application of the FAIR Principles for Findable, Accessible, Interoperable, and Reusable data publication ("The FAIR Data Principles – FORCE11," n.d.). These principles are a key driver in the ocean data market, as they are widely encouraged and even required by various organizations. The
authors provide recommendations for mitigating the challenges associated with the highly variable and dispersed nature of ocean data and suggest designing the global ocean data system as an interoperable system of systems that follows the FAIR principles through thematic integration of products and services. To achieve this goal, they suggest integrating existing data systems while enhancing their ability to digest and deliver data, thus allowing users easy access to diverse data (Tanhua et al., 2019b).

Buck et al. (2019) provide a key technological review and use cases of state-of-the-art ocean data systems, along with a vision and recommendations for the future. The authors introduce the concept of democratization of ocean data. They provide the vision of moving from a data portal model, by which users consume pre-built products, into a flexible data utilization model, where users can build knowledge systems based on interoperable ocean data services. They describe the ocean data life cycle, future workflows, standards, and the service-based architecture that is needed to support this approach.

Nativi et al. (2021) apply the Digital Ecosystem paradigm to describe the nature of a scalable (i.e., cloud-based) core platform designed to support Digital Earth or a high-precision digital model of the Earth. They continue to describe the digital ecosystem's technological framework and architecture. The paper identifies the characteristics of the digital ecosystem that are appropriate for connecting and orchestrating the many heterogeneous and autonomous online systems, infrastructures, and platforms.

Within the ocean data market, Pendleton et al. (2019) identify three classes of challenges to data sharing and use: uploading, aggregating, and navigating. The authors envision a disruptive data-sharing solution for the ocean data market aimed at helping data producers and users navigate the complexity of ocean data. They suggest technology platforms that combine aggregating and navigating technologies and social networks, similar to those recently applied to consumer products and the travel industry.

Vance et al. (2019) introduce cloud-based infrastructures in the context of the ocean observing system. This technology is suggested to support society and research needs, maximize the benefits of a more integrated ocean observing system, facilitate data and model sharing, support high-performance mass storage of observational data, and provide on-demand computing. The authors review topics of cloud-based scientific data such as getting and storing data from and in the cloud, computing infrastructure, and analyzing large datasets and datasets from multiple sources, and provide examples of

programs such as the NOAA Open Data Dissemination (NODD) Program ("Cloud Access | National Centers for Environmental Information (NCEI)," n.d.; Vance et al., 2019).

Brett et al. (2020) call for federated data networks to connect disparate ocean databases and for new incentives and business models for data sharing, with the aim of creating an "open, actionable, and equitable digital ecosystem for the sustainable future ocean."

The need for adopting the data ecosystem paradigm to facilitate the data sharing process in the marine domain has been demonstrated by Lima et al. (2022). Through data-centered discussions with participants from various organizations, the authors identified key challenges in ocean data sharing, such as technical limitations, legal liability, regulatory, privacy and security of data. They identify possible advantages from sharing marine data in the same ecosystem, thus emphasizing the benefits to society and to the organizations from adopting the data ecosystem approach (Lima et al., 2022).

**3. Modeling and mapping the ocean data ecosystem**

Following the above works, which have identified the domain of ocean data as a data ecosystem, and addressed some of its major constituents, here we propose a structured data ecosystem model that enables systematic and comprehensive mapping of the ocean data market landscape.

Our approach for mapping the ocean data ecosystem is inspired by the concept of "periodic table of elements of big data

value" proposed by Curry et al. (2021). The proposed model consists of 5 elements (Fig. 1): 1. "Stakeholders" (e.g. marine researchers and research institutions, regional and international ocean observing organizations and frameworks); 2. "Societal elements" (e.g. key principles, key initiatives, goals and targets, policy and regulation); 3. "Interoperability tools" (e.g. licenses, standards and marine ontologies, best practices and frameworks in ocean science); 4. "Data sources and product offering elements" (e.g. marine science data sources and their product and service offering; 4.5. "Emerging solutions" (e.g.

data integration solutions).

By developing a structured conceptual model of the ocean data ecosystem and providing illustrative examples, we intend to support readers in navigating this complex and evolving space. Rather than presenting an exhaustive and potentially quickly outdated inventory, the model is intended to help readers identify and characterize relevant examples (e.g. stakeholders, societal elements, integration tools, data sources and emerging solutions) that are most applicable to their specific domain,

use case, or geographic context. By presenting a flexible and structured framework, the model will serve as a tool to enable to add new developments and technologies, as they emerge in the ocean data ecosystem.

For the purpose of providing an interactive map, we created an online ocean data ecosystem relational model available at https://kumu.io/odini/ocean-data-ecosystem the map is a long-term reference, open, and may be updated and extended. To facilitate contributions and comments from the public, we make a public issue system available https://gitlab.com/odini_dev/data-ecosystem-ontology) and invite readers to suggest additions and corrections. The methodology we used is based on a thorough literature and website review of over 90 scientific articles and over 100 websites. Articles and examples selected to illustrate the elements of the model have been selected based on using search terms such as 'ocean data,' 'ocean data interoperability,' and 'marine ontologies.' The examples are not exhaustive and are intended as a starting point for ocean data professionals, who are encouraged to explore additional resources specific to their research areas.

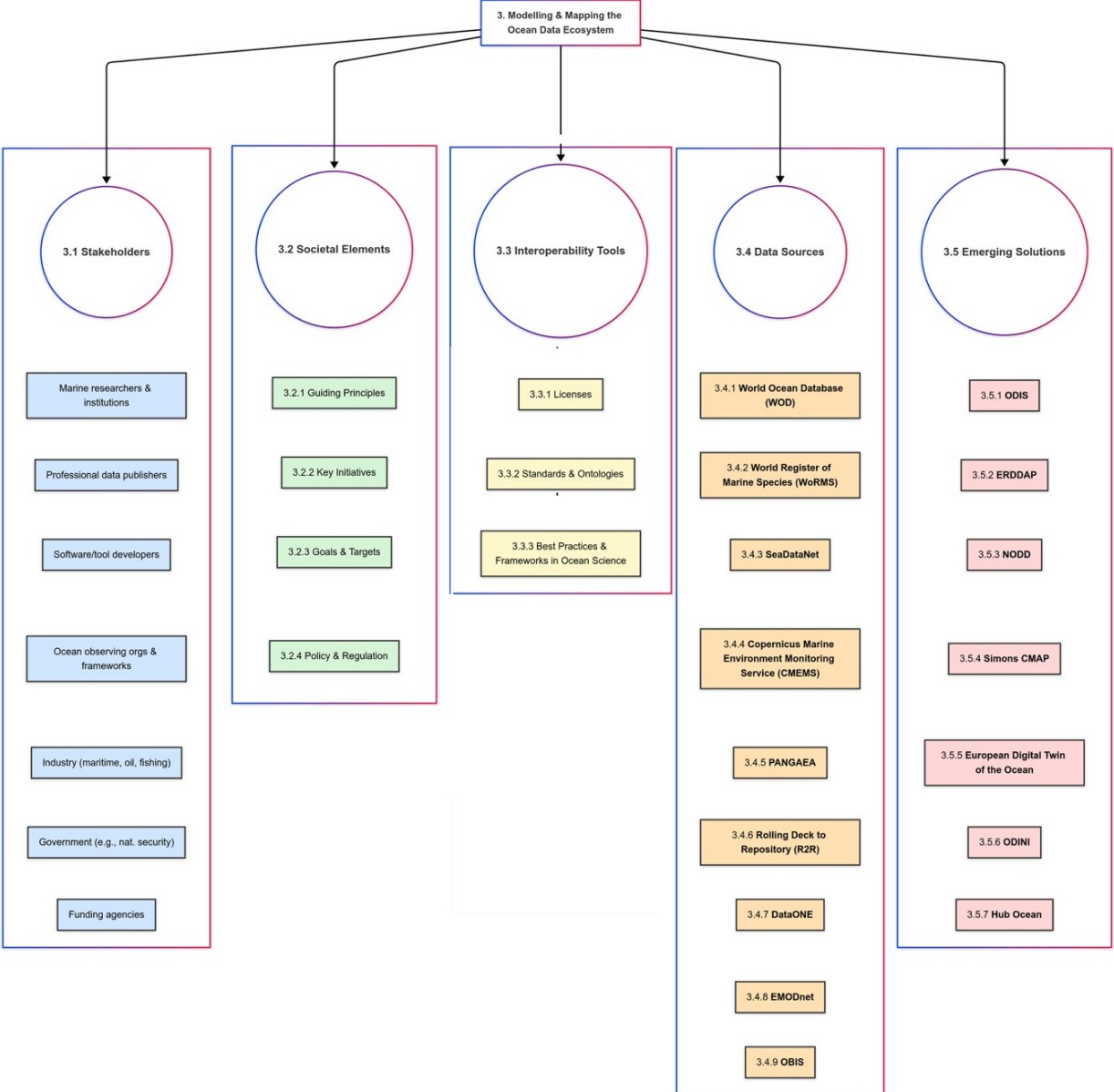

**Figure 1.** A diagram showing the main actors in our ocean data ecosystem model. The section, sub-section and illustration example numbers to help the reader orient within each section.

We now elaborate on these different elements, followed by a discussion on the different roles they play in the ocean data ecosystem.

### 3.1. Stakeholders

The main stakeholders of the ocean data ecosystem were described by Tanhua et al. (2019a). These include the researchers, professional data publishers, software and tool builders, funding agencies and the data science community. To facilitate mapping the oceans data landscape, in our model the stakeholders are categorized according to the following groups:

- Marine researchers and research institutions: The marine researcher is our focus as the data end-user. Other players may be software developers, marine device technology developers, users in the marine industry, funding agencies, users in national and international organizations and citizens. Other examples of data end users may be professional data publishers, software tool developers.

- Regional and international ocean observing organizations and frameworks (data producers): for example: The Global Ocean Observing System (GOOS) (Capet et al., 2020; "Global Ocean Observing System," n.d.; Tanhua et al., 2019a), which provides countries and end-users with critical information on physical, chemical, and biological essential ocean variables, European Ocean Observing System Framework (EOOS) ("European Ocean Observing System," n.d.), the US Integrated Ocean Observing System (IOOS) ("The U.S. Integrated Ocean Observing System (IOOS)," n.d.), the Joint Technical Commission of Oceanography and Marine Meteorology in situ Observing Platform Support (JCOMMOPS) ("OceanOPS," n.d.) and many others. In their activities, ocean observing frameworks and organizations are also key drivers of the ecosystem.

- Industry (maritime, oil, fishing), government (national security) and funding agencies (data producers and data end users): These and other stakeholders were not part of the focus of our analysis, which focused mainly on ocean data for the purpose of answering oceanographic research questions.

### 3.2. Societal elements

Like other data ecosystem models (Curry et al., 2021), the societal elements of the ocean data ecosystem refer to components that play a part in the societal, regulatory, organizational and technological context. In our model, this includes four components (Fig. 2): 1. guiding principles, 2. key initiatives, goals and targets, 3., and 4. policy and regulations.

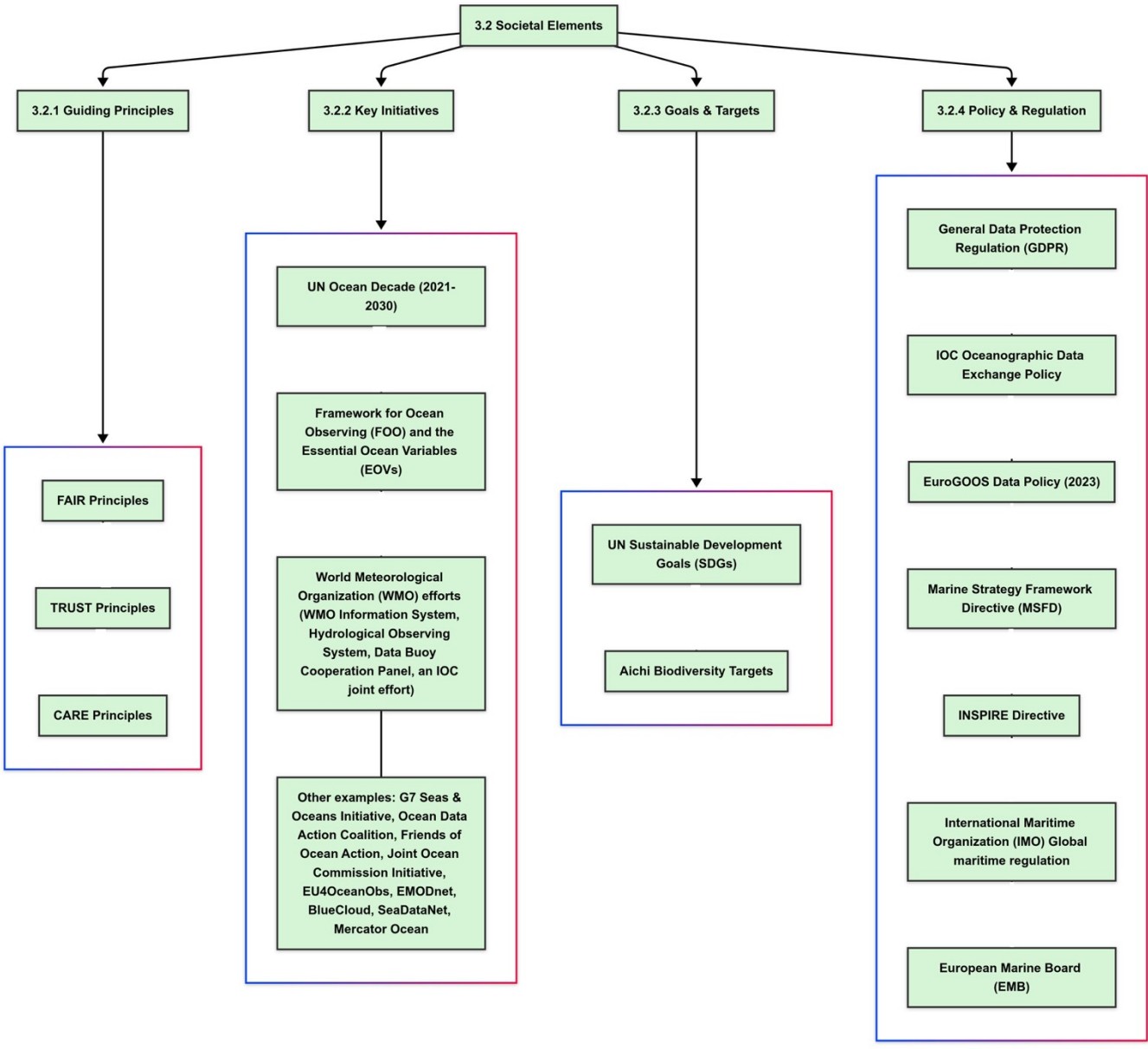

**Figure 2.** A diagram summarizing the types of societal elements, and the representative examples discussed in Sec. 3.2.


### 3.2.1. Guiding principles

By and large, the ocean data management is guided by the FAIR, TRUST and CARE principles, each is designed to serve a different purpose and applicable in different contexts. The FAIR principles are mainly concerned with scientific data management and stewardship and are meant to ensure reusability of data, with an emphasis on enabling the automation of

data findability and usability (Tanhua et al., 2019b; "The FAIR Data Principles – FORCE11," n.d.; Wilkinson et al., 2016). The TRUST principles of data repositories (Lin et al., 2020), deals with data curation, providing guidance to demonstrate transparency, responsibility, user focus, sustainability, and technology. The CARE principles ("CARE Principles - Global Indigenous Data Alliance," n.d.) were defined for ensuring proper handling of data associated with indigenous communities, defining measures for Collective Benefit, Authority to Control, Responsibility, and Ethics.

In our context of modeling the ocean data ecosystem with a focus on data interoperability, we find the FAIR principles a useful tool, as they are meant to provide data producers and publishers measurable guidelines for ensuring their data implementation to be Findable, Accessible, Interoperable, and Reusable, to overcome the barriers for large scale data utilization. The need to follow the FAIR principles stems from the fact that gathering the data required for answering research questions is often a tedious and time-consuming task, largely due to lack of attention to how the data assets are
preserved when they are created (Tanhua et al., 2019b). The FAIR principles answer to the step-by-step process by which machines will be able to process the data, identifying the relevant data within a given context, determining if it is useful, if it is usable in terms of license or other accessibility and taking action (Tanhua et al., 2019b).

### 3.2.2. Key initiatives

Key initiatives refer to coordinated efforts designed to promote effective data management, sharing, and stewardship within the data ecosystem. On a global scale, a pivotal initiative within the ocean data ecosystem is the UN Ocean Decade, which was proclaimed in 2017 by the United Nations General Assembly, the UN Decade of Ocean Science for Sustainable Development (2021-2030) ("Ocean Decade – The Science We Need For The Ocean We Want," n.d.; Ryabinin et al., 2019). The UN Ocean Decade serves as a framework for scientists and stakeholders to establish partnerships and develop solutions
for improving our understanding of oceanic systems and promoting science-based decision making. Being a pivotal initiative within the ocean data ecosystem, the UN Ocean Decade has been the driving force for a number of important ocean data ecosystem components. The coordination between the different components is done through the Intergovernmental Oceanographic Commission of UNESCO (IOC), ("Intergovernmental Oceanographic Commission | Intergovernmental Oceanographic Commission," n.d.), which aims at generating ocean knowledge and promoting international cooperation by
leveraging a global network of experts, scientists, and partners. Within the IOC, oceanographic data and information exchange is facilitated by the International Oceanographic Data and Information Exchange program (IODE) ("IODE – International Oceanographic Data and Information Exchange," n.d.), which is global network of more than 100 National Oceanographic Data Centres (NODCs), Associate Data Units (ADUs) and Associate Information Units (AIUs). Other key components of IOC include The Global Ocean Data and Information System (ODIS) ("Ocean Data Information System,"
n.d.), which is a partnership of independent systems that aims at leveraging existing  solutions, through sharing metadata and

information; The Ocean InfoHub project ("Ocean Infohub," n.d.), which was a three-year project initiated on 2020 in order to promote a sustainable, interoperable, and inclusive digital ecosystem for all ocean stakeholders.

The IOC Ocean and Data Information system Catalog of data sources (ODISCAT) ("IOC Ocean Data and Information System Catalogue," n.d.), which is an online catalog of existing ocean related web-based sources/systems of data, information, products and services, currently describing over 1000 online sources of marine and coastal data in 16 categories.

Another key initiative with the ocean data ecosystem is the Framework for Ocean Observing (FOO, deYoung et al., 2019; Lindstrom et al., 2012; Tanhua et al., 2019a), which was developed by a task team of the ocean observing community through sponsorship of IOC, with implementation coordinated by the GOOS. The FOO serves as a guideline for developing a multidisciplinary, integrated ocean observing system for operational purposes, and uses the Essential Ocean Variables (EOV) as guideline. Other key initiatives include the World Meteorological Organization (WMO) ("World Meteorological Organization WMO," n.d.) and its technical framework the WMO Information System ("WMO Information System (WIS)," n.d.) the WMO Hydrological Observing System ("WMO Hydrological Observing System (WHOS)," n.d.), and the Data Buoy Cooperation Panel ("DBCP Data Buoy Cooperation Panel," n.d.), which is an official joint body of the World Meteorological Organization (WMO) and the Intergovernmental Oceanographic Commission (IOC); Ocean Action 2030 ("Ocean Action 2030 - Ocean Panel," n.d.) and its derivatives - the Ocean Data Action Coalition (ODAC) ("The Ocean Data Action Coalition - HUB Ocean | Dedicated to Unlocking Ocean Data," n.d.) and the Friends of Ocean Action ("FRIENDS OF OCEAN ACTION > Friends of Ocean Action | World Economic Forum," n.d.); G7 Future of the Seas and Oceans Initiative ("G7 Future of Seas and Ocean Initiative," n.d.), Joint Ocean Commission Initiative (JOCI) ("Joint Ocean Commission Initiative," n.d.); JPI Oceans ("JPI Oceans," n.d.), Mercator Ocean International ("Mercator Ocean - Ocean Forecasters," n.d.); EU4OceanObs ("Use Ocean Data & Information - EU4OceanObs," n.d.), which aims to enhance the uptake of EU ocean data sharing initiatives and their applications, by creating synergies with the activities of Copernicus and its Marine and Climate Services and other EU ocean data and sharing infrastructures and portals such as EMODnet ("European Marine Observation and Data Network (EMODnet)," n.d.; Míguez et al., 2019), BlueCloud, and SeaDataNet; and OceanOPS ("OceanOPS," n.d.), which provide integrated information, maps and tools on global ocean observation efforts.

### 3.2.3. Goals and targets

Goals and targets refer to large-scale strategic objectives which are defined by pivotal organizations and international agreements. Goals and targets can have a very broad perspective, as for the case of the UN Sustainable Development Goals ("THE 17 GOALS | Sustainable Development," n.d.), which consists of 17 interconnected global objectives established by the United Nations to address critical challenges such as poverty, inequality, climate change, environmental degradation,

peace, and justice. Another example for global-scale broadly accepted objectives can be found the AICHI targets for managing biodiversity ("Aichi Biodiversity Targets," n.d.), which, among other things, calls for integration of biodiversity
data sets from a range of disparate sources (Buck et al., 2019). Ocean data supports the achievement of such large-scale common goals by providing the necessary information to make informed decisions, develop policies, and implement effective management strategies.

### 3.2.4. Policy and regulation

Policy and regulation in the field of ocean data refers to the established guidelines and legal frameworks that govern data ownership, usage, protection, privacy, security, and exchange, to ensure that oceanographic data are managed responsibly and sustainably, while safeguarding individual rights and promoting open data access. Examples include The General Data Protection Regulation ("General Data Protection Regulation (GDPR) – Legal Text," n.d.) implemented by the European Union, which sets strict guidelines for data ownership, usage, protection, and privacy, ensuring that personal data collected
and processed are securely handled and that individuals' privacy rights are upheld; The IOC Oceanographic Data Exchange Policy  ("Intergovernmental Oceanographic Commission | Intergovernmental Oceanographic Commission," n.d.) that promotes the free and open exchange of oceanographic data; The EuroGOOS Data Policy (2023), which provides recommendations for incorporation of data management plans that ensures the adoption of FAIR principles from the early stages of data production. The Marine Strategy Framework Directive (MSFD) ("MSFD," n.d.), which is an EU directive,
which requires member states to monitor and assess the environmental status of their marine waters;  The Infrastructure for Spatial Information in Europe Directive (INSPIRE) ("INSPIRE Knowledge base - European Commission," n.d.), which is an EU directive that aims to enable the sharing of environmental spatial information among public sector organizations and better facilitates public access to spatial information across Europe; The International Maritime Organization (IMO) ("International Maritime Organization," n.d.), which develops and maintains a comprehensive regulatory framework for
different aspects of shipping activity; and the European Marine Board (EMB) ("European Marine Board," n.d.) which aims to develop marine research foresight and initiate state-of-the-art analyses that can used for policy recommendations to European institutions and governments on national and international levels.

### 3.3. Interoperability tools and frameworks

Interoperability, the ability of different systems and data-driven solutions to work together seamlessly, is crucial in supporting the vision of a distributed ocean data ecosystem (Buck et al., 2019; Curry et al., 2021; Pearlman et al., 2021, 2016; Tanhua et al., 2019a). It enables efficient data exchange, integration, discoverability and analysis across diverse platforms and sources. Over the past decade, the marine domain has witnessed significant evolution in interoperability, driven by stakeholder collaboration and emerging needs. As ocean data sources and platforms evolved, and the need to

reduce manual work and streamline the data value chain has become evident (Sagi et al., 2020), the community developed interoperability tools. Interoperability tools such as standards, vocabularies, and protocols have been defined and implemented to ensure adherence to FAIR principles, with a particular emphasis on facilitating machine-readable and machine-actionable data (Tanhua et al., 2019b), and definition and adoption of best practices and frameworks. The interoperability tools and framework elements of the ocean data ecosystem refer to the various ways by which stakeholders address the interoperability challenges facing the ocean data ecosystem (Fig. 3). The major approaches taken to address these challenges include 1. Licenses and accreditations for marine data management, 2. Best Practices and Frameworks, and 3. standards, vocabularies and ontologies). We now give an overview on these components.

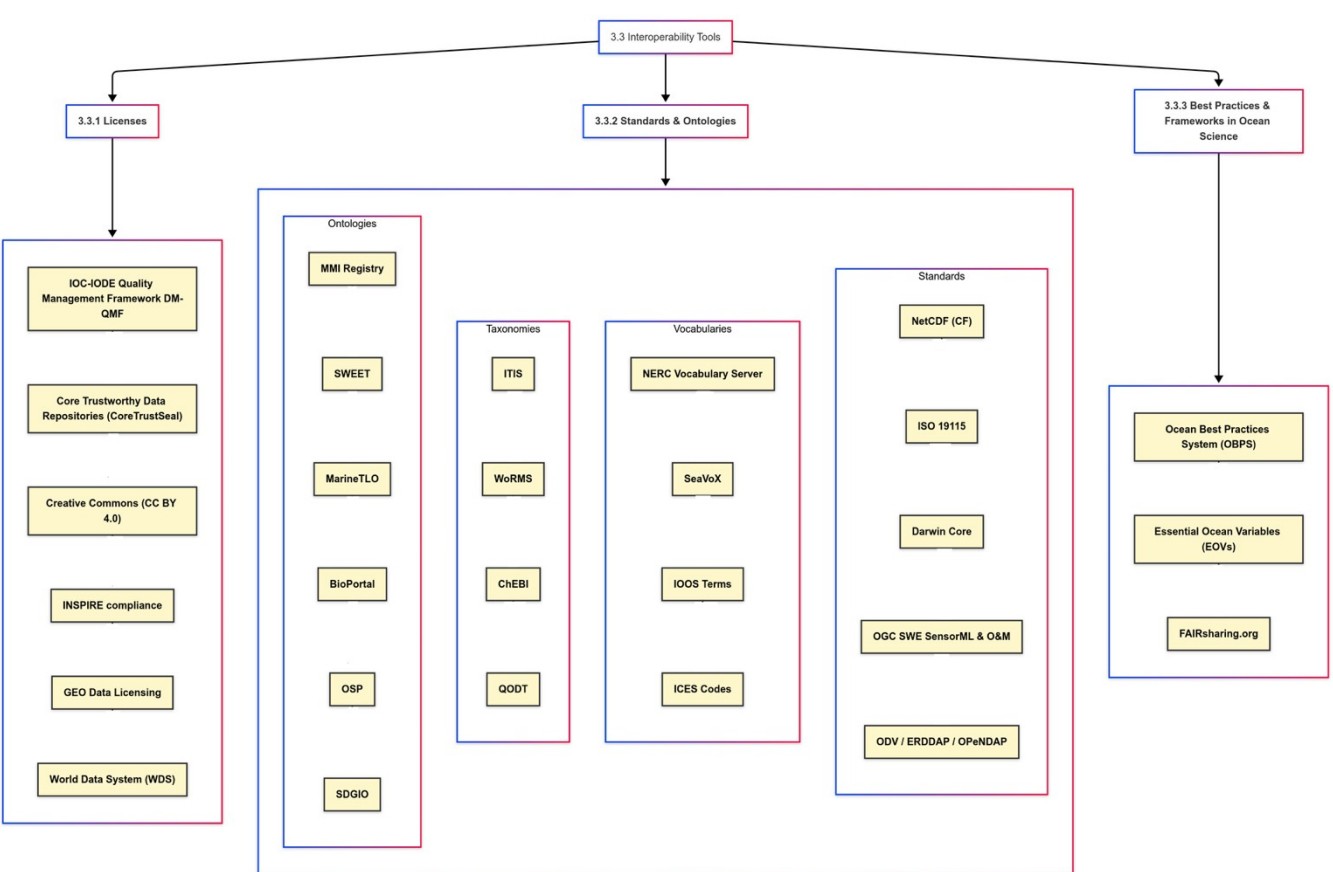

**Figure 3.** A diagram summarizing the types of interoperability tools and the representative examples discussed in Sec. 3.3.

### 3.3.1 Licenses and accreditations for marine data management

Accreditations for ocean data management provides a roadmap and guidelines for data management systems and helps to
build stakeholder confidence in data processes. In addition, embracing such data management accreditations improves the quality and transparency of data processes and management of 3rd party data ("Proceedings Volume International Conference on Marine Data and Information Systems IMDIS 2024 - Bergen (Norway), 27-29 May 2024," n.d.).

Examples for Licenses and Accreditations include:

The IOC-IODE Quality Management Framework (DM-QMF) ("IODE quality management framework for national
oceanographic data centres and associate data units - UNESCO Digital Library," n.d.; Leadbetter et al., 2020), which was developed to assist National Oceanographic Data Centres (NODC) ("IODE – International Oceanographic Data and Information Exchange," n.d.) network to establish organizational data management quality management tools. The IOC-IODE's framework also promotes the accreditation of NODCs which have implemented adhering to the guidelines laid out in the IOC-IODE's framework. Leadbetter et al. (2020) provide an example from the Marine Institute of Ireland, which also
includes helpful templates.

CoreTrustSeal ("CoreTrustSeal – Core Trustworthy Data Repositories," n.d.) is an international, community-based, non-governmental, and non-profit organization that aims to promote sustainable and trustworthy data infrastructures. This is done by issuing a CoreTrustSeal certification, which offers to any interested data repository a core level certification based on its requirements.

The Creative Commons (CC) ("Open Science - Creative Commons," n.d.)**,** is an international non-profit organization which promotes open sharing of data facilitated using standard, public legal tools used to manage copyright and similar restrictions that might otherwise limit dissemination or reuse of data. In the CC-BY license, credit must be given to the creator. This license is recommended by EuroGOOS ("EuroGOOS Data Policy 2023.," 2023). The ICES International Council for the Exploration of the Sea (ICES) ("ICES," n.d.) data policy, maximizing the availability of data to the community by ensuring
all public data are under the Creative Commons (CC BY 4.0) license.

Other examples include INSPIRE ("INSPIRE Knowledge base - European Commission," n.d.). (e.g. SeaDataNet is achieving INSPIRE compliance for some metadata services directory, (Pecci et al., 2020). The GEO Data Licensing Guidance ("Data Licensing Guidance," n.d.); The World Data System by the International Council for Science (ICS) ("World Data System," n.d.).


### 3.3.2. Standards, vocabularies and marine ontologies

An important step towards data interoperability is the formations *standards,* which are sets of guidelines, specifications, accepted practices, technical requirements, or terminologies that are documented and agreed-upon by the research

community ("How do we define standards?," n.d.). Standards are complemented by vocabularies (that are lists of terms relevant to the research domain) and ontologies (that describe the relationships between the terms), which are used by the research community to describe metadata and datasets (Tanhua et al., 2019b). Here we give a brief overview on some of the available Standards, Vocabularies and Ontologies and other interoperability tools used in the ocean data ecosystem (Buck et al., 2019; Carbotte et al., 2022; Felden et al., 2023a; Hankin et al., 2010; Míguez et al., 2019; Tanhua et al., 2019b).

A detailed description of major data and metadata standards and relevant tools is provided in Tables 1 and 2 of Buck et al (2019). The NetCDF ("CF Conventions," n.d.), which is metadata conventions for describing Climate and Forecast (CF) data (Buck et al., 2019) and the adapted SeaDataNet NetCDF CF import format ("Data Transport Formats - SeaDataNet," n.d.). NCEI NetCDF templates ("NetCDF Templates | National Centers for Environmental Information (NCEI)," n.d.) assist data producers to conform to CF conventions (Buck et al., 2019; Tanhua et al., 2019b). The CDI (Climate Data Interface) Data Access Interface is part of the CDI library, which is a software toolset developed for accessing and manipulating climate data ("CDI - CDI - Project Management Service," n.d.).

Geographic data standards include the ISO 19115 ("ISO 19115-1:2014 - Geographic information - Metadata - Part 1: Fundamentals," n.d.), an international standard for describing geographic metadata and is used, for example, by PANGAEA (Felden et al., 2023a). The Open Geospatial Consortium (OGC) develops standards ("Standards - Open Geospatial Consortium," n.d.) focused on making geospatial and location-based data interoperable across different systems. EMODnet Chemistry utilizes OGC-compliant formats in its various product offerings (Míguez et al., 2019). The National Marine Electronics Association (NMEA) formats ("Standards - National Marine Electronics Association," n.d.), is a standard data format supported by GPS manufacturers.

The Darwin Core is a globally accepted standard for biodiversity information ("What is Darwin Core, and why does it matter?," n.d.; Wieczorek et al., 2012), and is supported, for example by PANGAEA and EMODNet Biology (Felden et al., 2023a; Míguez et al., 2019).

DataCite DOIs ("Introduction to the DataCite REST API," n.d.) provide persistent unique identifiers, making data citable, and is being used, among others, by the Rolling Deck to Repository program (Carbotte et al., 2022; "Rolling Deck to Repository (R2R)," n.d.) for cruise metadata records, and by EMODnet Chemistry (Míguez et al., 2019);

OGC's Sensor Web Enablement (SWE) standards, including SensorML and Observations and Measurements (O&M), ensure consistent data representation (Buck et al., 2019). The OPeNDAP data access protocol contributes to interoperability of ocean data by providing a standardized protocol that allows users to access and retrieve remote and large datasets regardless of the storage format (Buck et al., 2019; Hankin et al., 2010). The ERDDAP data server ("ERDDAP," n.d.) is aimed at providing a consistent way to download subsets of scientific datasets in common file formats, including oceanographic data, used for example by Argo (Buck et al., 2019).

Standards for interoperability of visualization of oceanographic datasets includes the Ocean Data View ("ODV: ODV," n.d.), which is a free ocean data visualization, analysis and manipulation tool for large environmental datasets. The SeaDataNet ("SeaDataNet - SeaDataNet," n.d.) has adopted the ODV as its data analysis and visualization software and requires its datasets to support the SeaDataNet ODV4 ASCII format ("Data Transport Formats - SeaDataNet," n.d.). SeaView, an EarthCube project ("Products | EarthCube," n.d.; "SeaView Data," n.d.), is working with existing data repositories and aimed at interoperability of oceanographic data, through partnership with actors such as The Biological and Chemical Oceanography Data Management Office (BCO-DMO) ("Introduction to BCO-DMO | BCO-DMO," n.d.), The Ocean Biodiversity Information System (OBIS) ("Ocean Biodiversity Information System," n.d.) and the Rolling Deck to Repository program ("Rolling Deck to Repository (R2R)," n.d.).

Methods enabling the discoverability of datasets include schema.org ("Schema.org - Schema.org," n.d.), which is a standard for structured knowledge about data, created through a collaborative effort by major search engines such as Google. Google dataset search ("Dataset Search," n.d.) utilizes schema.org to help users discover datasets on the web. Web-accessible metadata and schema.org protocols enhance dataset search and discovery. For example, Rolling Deck to Repository Data sets (R2R) can be discovered through Google dataset searches and through the EarthCube GeoCODES portal (Carbotte et al., 2022; "GeoCodes," n.d.).

Commonly used vocabularies in the ocean data ecosystem, which can be defined as lists of standardized terms from a wide array of oceanographic disciplines, can be found in the IOOS ontologies, common vocabularies, and ("Ontologies, Common Vocabularies, and Identifiers - The U.S. Integrated Ocean Observing System (IOOS)," n.d.), and the NERC Vocabulary Server ("NERC Vocabulary Server," n.d.; "NVS," n.d.)

SeaDataNet has been involved in developing standards and vocabularies, provides a library of vocabularies, and a directory of over 4000 marine research organizations ("Common Vocabularies - SeaDataNet," n.d.).

The International Council for the Exploration of the Sea (ICES) vessel vocabulary ("ICES Reference Codes - RECO," n.d.), provides internationally agreed upon controlled vocabularies for cruise metadata (e.g. vessels), which are used, for example, by The Rolling Deck to Repository program (Carbotte et al., 2022; "Rolling Deck to Repository (R2R)," n.d.). The SeaVoX Device Catalog is a device type vocabulary, which is hosted by the British Oceanographic Data Center and implemented, for example, in the SeaDataNet system (Pecci et al., 2020; Schaap and Lowry, 2010; "SeaDataNet - SeaDataNet," n.d.).

Ontologies are used to describe the relationships between entities in various domains of ocean research. Ontologies provide a formal specification of concepts and relationships, enhancing data integration, but a complete oceanographic ontology has not yet been constructed. Examples for ontologies used in the ocean data ecosystem include the MMI Ontology Registry and Repository ("Marine Metadata Interoperability Project Semantic Web Services," n.d.; Rueda et al., 2009) that is an online repository for marine ontologies; The Integrated Ocean Observing System ("Ontologies, Common Vocabularies, and Identifiers - The U.S. Integrated Ocean Observing System (IOOS)," n.d.) that is responsible, for example to the IOOS

Biological Data Ontology ("GitHub - ioos/vocabularies: Instructions and Guidelines for use of Controlled Vocabularies in IOOS-compliant data services," n.d.); The Semantic Web for Earth and Environment Technology Ontology (SWEET) (Raskin and Pan, 2005; "Semantic Web for Earth and Environment Technology Ontology | NCBO BioPortal," n.d.); The Marine Top-Level Ontology for the marine domain ("MarineTLO | A Top Level Ontology for the Marine/Biodiversity Domain," n.d.; Tzitzikas et al., 2016); NERC Ontologies and Ontology Extension for Marine Environmental Information Systems (Leadbetter et al., 2014; "Overview | NETMAR," n.d.); Marine Regions Gazetteer Ontology (Lonneville et al., 2021; "Marine Regions," n.d.), which is a standard list of marine georeferenced place names and areas. The EUCISE-OWL ("Ontologies - EU Vocabularies - Publications Office of the EU," n.d.; Riga et al., 2021) is an ontology-based system designed to support the European Common Information Sharing Environment (CISE) for the maritime domain, aiming to make existing maritime data systems more interoperable. The BioPortal web application of the National Center for Biomedical Ontology (Fergerson et al., 2015; "NCBO BioPortal," n.d.) is a repository where users can search a library of more than 1,300 biomedical ontologies (as of December 2024).

Other recent marine ontology include the OSP Maritime Domain Ontology ("OSP Maritime Domain Ontology - Open Simulation Platform," n.d.; Troupiotis-Kapeliaris et al., 2022); Semantic Sensor Network Ontology (SSN/SOSA) ("Semantic Sensor Network Ontology," n.d.), Ocean Circulation Spatial–Temporal Ontology (Zhang et al., 2023); Autonomous Vessel Design (Arrigan et al., 2022). GeoReservoir ontology for deep-marine depositional system geometry description (Cicconeto et al., 2022); Climate change (Surya et al., 2021); BiGe-Onto for managing biodiversity and biogeography data. (Zárate et al., 2020); Oceanic Data Description Extraction Project ("OSF | Oceanic Data Description Extraction Project," n.d.). The GeoLink Dataset ("EarthCube GeoLink," n.d.; Zhou et al., 2018; Zhuang et al., 2016); D-Ocean; The UN Environment Sustainable Development Goals Interface Ontology (SDGIO) (Buttigieg et al., 2016).

Additional interoperability tools in the ocean data ecosystem include the Integrated Taxonomic Information System (ITIS) ("Integrated Taxonomic Information System," n.d.) and World Register of Marine Species ("WoRMS - World Register of Marine Species," n.d.) taxonomy terminologies, Chemical Entities of Biological Interest ("Chemical Entities of Biological Interest (ChEBI)," n.d.) taxonomy, the QUDT ("QUDT," n.d.) measurements taxonomy. These are utilized for example by PANGAEA ("Data Publisher for Earth & Environmental Science," n.d.; Felden et al., 2023b).

### 3.3.3. Best practices and frameworks
The notion of ocean best practices refers to a set of methodologies and workflows across ocean research, which were found to provide improved (with respect to other methodologies) results, making them broadly adopted by various organizations. The Ocean Best Practices System (OBPS) (Buttigieg et al., 2019; Hörstmann et al., 2021; "Ocean Best Practices System," n.d.; Pearlman et al., 2021, 2017), describes key standardization and best practice activities within the ocean data ecosystem, driving adoption of common data solutions and services within the ecosystem.

The FOO uses Essential Ocean Variables (EOV) (deYoung et al., 2019; "Essential Ocean Variables – Global Ocean

Observing System," n.d.) as a framework to determine the key observations that are required to achieve the goals of the observing system. The EOVs for physics, biogeochemistry, and biology/ecosystems are negotiated based on feasibility and impact and are used by ocean data management systems to prioritize measurements that should be made.

Other resources on data and metadata standards can be found at websites such as FAIRsharing ("FAIRsharing," n.d.), which help to discover and use resources related to databases and data policies.


### 3.4. Data sources and product offering

Ocean data sources and their product offering serve as the infrastructural elements of the ocean data ecosystem, where data producers and data users interact. This element hosts the main asset of the ecosystem, the ocean data, and provides the access point for the users, making it the heart of our ocean data ecosystem model. Following Oliveira et al. (2019), data sources can

be defined as platform-centric structures that provide infrastructures and services to support both the provision and consumption of data. These infrastructural components are major building blocks of the distributed system of systems (Tanhua et al., 2019b).

By and large, the various data sources can roughly be divided into three types: *Raw sources, repositories and portals*. Data are collected by devices and expeditions that are part of numerous scientific projects and ongoing efforts. Many of these

efforts sustain their own websites where they publish the data periodically or as a live stream. In our model these are called *Raw Sources*. *Repositories* store data from multiple sources that use these repositories to archive their data and make them more publicly available, Pangaea is an example of a data repository ("Data Publisher for Earth & Environmental Science," n.d.; Felden et al., 2023a). *Portals* provide an interface to search in multiple repositories but do not host the data themselves. For example, DataONE ("Data Observation Network for Earth | DataONE," n.d.; Michener et al., 2011), is a portal through

which one can search multiple repositories and find project data, for example data hosted on PANGAEA.

Notably, different data sources may exhibit significant similarity, often with overlap between their contents. In general, each data source serves a distinct user community, and datasets may be duplicated throughout the ecosystem. For example, many records in the World Ocean Database (WOD, see below Sec. 3.4.1) are compiled from subsets of data originating in datasets such as PANGAEA (see below Sec. 3.4.5.), but are curated into a product designed for a specific purpose and audience.

Overlap and similarity between data sources can be exemplified for the cases of SeaDataNet (see below Sec. 3.4.3) and DataONE (see below Sec. 3.4.7), which although differing in their funding structures, partnership frameworks and infrastructural approach, align in their high-level functions for end users.

Here we categorize the different data sources by their infrastructural approach, as defined in section 2.1, namely: *centralized, federated,* and *distributed* data ecosystems.

The description of data sources, tools, and frameworks is not exhaustive. Due to the very broad nature of the ocean data ecosystem, rather than covering the large number of elements it contains, we give several representative examples. The readers may use these examples, presented in detail, to broaden their knowledge of the data ecosystem, and may perform further review of other data sources within their respective fields. We used a framework for analyzing the data sources, including key characteristics composing a data source, namely: Organization details such as number of partnering

organizations, oceanographic domain, geographic region, number of data sets, data catalogue and product offering, main uses by the specific user community and interoperability strategy. Other characteristics may be selected depending on the reader's focus and interests.

We now give an overview on some of the major data sources (Fig. 4).

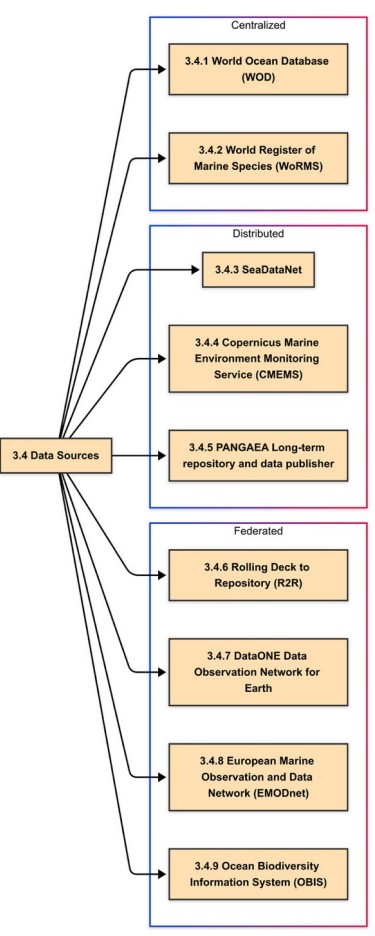

**Figure 4.** A diagram summarizing the examples for ocean data sources discussed in section 3.4. The exemplified data sources are organized according to their infrastructural approach (namely federated, distributed and centralized).

### 3.4.1. World Ocean Database (WOD)

*Data source infrastructural approach: Centralized data management*

The WOD (Levitus et al., 2013; "World Ocean Database | National Centers for Environmental Information (NCEI)," n.d.), as well as the World Ocean Atlas ("World Ocean Atlas | National Centers for Environmental Information (NCEI)," n.d.) that is derived from it, are maintained by NOAA's National Centers for Environmental Information ("National Centers for Environmental Information (NCEI)," n.d.). The WOD is an International Oceanographic Data and Information Exchange (IODE) project, with a mirror site hosted at the IODE ("World Ocean Database Select and Search," n.d.) and with major

releases and quarterly updates. Consisting of more than 20,000 data sets, over 15.7 M oceanographic casts and 3.6 B individual profile measurements taken by approximately 800 institutes around the world, the WOD is one of the largest and most comprehensive collections of oceanic data incorporated into a single database that is freely available to the public. The WOD data are commonly used for various oceanographic implications, such as creating boundary conditions for ocean models, and tracking changes in the state of the World Ocean over decadal, annual, seasonal, and monthly time scales. Fig. 5

shows the Water Column Sonar Data Viewer, as an example product of the World Ocean Database.

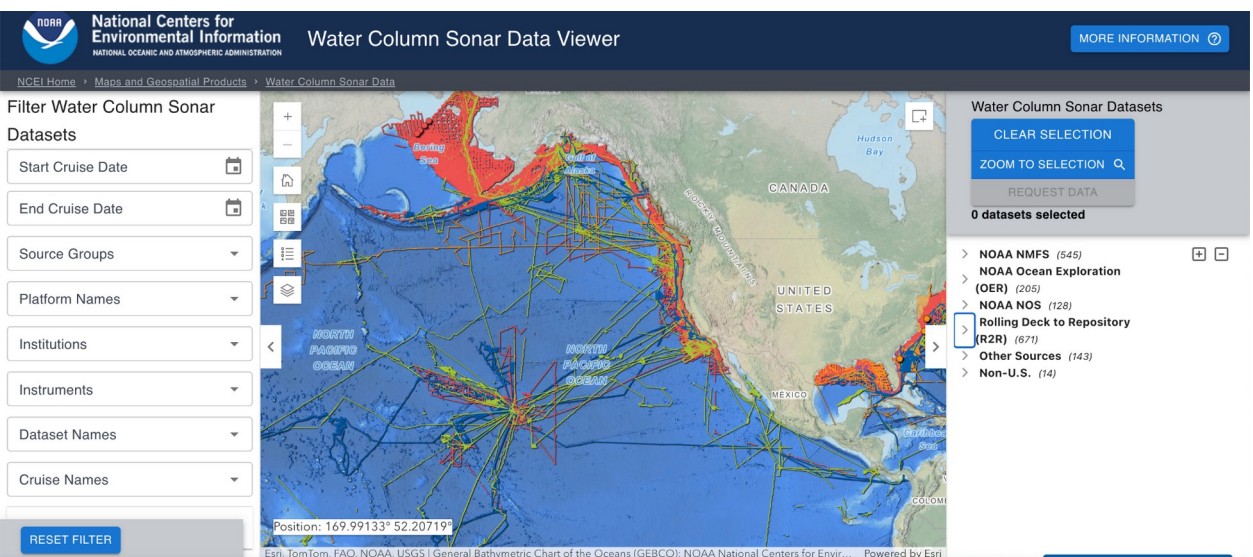

**Figure 5.** The Water Column Sonar Data Viewer, an example product from the World Ocean Database. (Screenshot from the NOAA National Centers for Environmental Information (NCEI) Water Column Sonar Data Viewer website 530 (https://www.ncei.noaa.gov/maps/water-column-sonar/), accessed on 26 December 2024. Public domain.) ("Water Column Sonar Data Viewer," n.d.).

**Interoperability strategy**. The WOD comprises quality controlled and uniformly formatted data from various sources that are incorporated into a single database, grouping together data acquired in a similar manner. The WOD can be searched by

specific parameters (e.g. date, geographic area) and measured variables. Download formats include WOD native, csv, or NetCDF ("World Ocean Database | National Centers for Environmental Information (NCEI)," n.d.)

### 3.4.2. World Register of Marine Species (WoRMS)

*Data source infrastructural approach: Centralized data management*

World Register of Marine Species ("WoRMS - World Register of Marine Species," n.d.) is a global effort of over 180 institutions from 38 countries, which aims to register all marine species names, including information on synonymy, in a quality controlled consolidated database. The content of WoRMS is controlled by taxonomic and thematic experts. Statistics are provided via the website, as for August 2024 the database consists of over 1 M records, over 600,000 checked biota names, with usage of over 2.5 M unique visitors in 2023.

### 3.4.3. SeaDataNet

*Data source infrastructural approach: Distributed marine data management*

    SeaDataNet (Pecci et al., 2020; Schaap and Lowry, 2010; "SeaDataNet - SeaDataNet," n.d.), is a framework for data sharing and collaboration, comprising organizations as the IOC-IODE and the ICES. The framework provides a distributed marine data infrastructure for the management of a variety of marine datasets, at European and Global scales, gathering data from
over 100 data centers, 35 countries and 2 million datasets. SeaDataNet provides a single website access to multidisciplinary ocean data, data products (Fig. 6), a data catalog, metadata services and software tools for data analysis. Stakeholders include members of the marine research community, government agencies, industry, and the general public. SeaDataNet provides quality checked data products for six European marine basins (Arctic Sea, Baltic Sea, Black Sea, Mediterranean Sea, North Sea, and North Atlantic Ocean) ("Products - SeaDataNet," n.d.), and regional climatologies based on the
aggregated datasets from external data sources such as Coriolis Ocean Dataset for Reanalysis ("CORA - Coriolis : In situ data for operational oceanography," n.d.) and the World Ocean Database (WOD) ("World Ocean Database | National Centers for Environmental Information (NCEI)," n.d.).

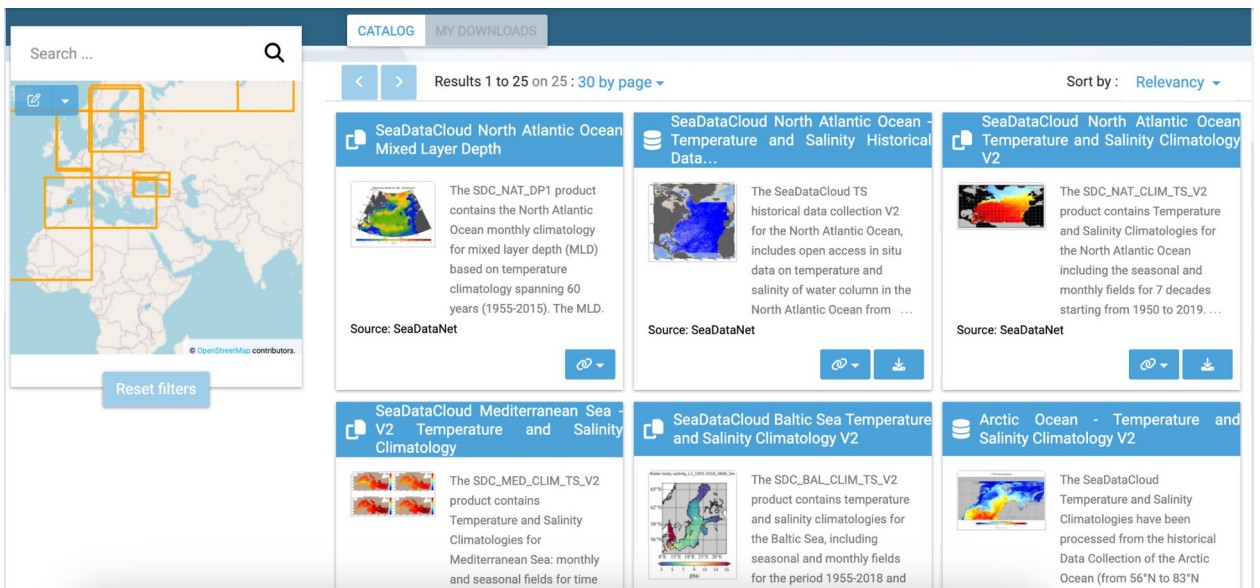

**Figure 6.** SeaDataNet data products from aggregated datasets (Screenshot from the SeaDataNet website (https://www.seadatanet.org/Products), accessed on 26 December 2024. Used under the Creative Commons Attribution 4.0 International (CC-BY-4.0) license ("Products - SeaDataNet," n.d.).

Another useful service provided by SeaDataNet is the European directory of marine research organizations ("EDMO -

Organisations - SeaDataNet," n.d.), which describes more than 4,000 organizations engaged in oceanographic and marine research activities, data and information management and data acquisition activities.

**Interoperability strategy.** A major objective and challenge in SeaDataNet are to provide an integrated and harmonized access to data resources, using a distributed network approach. This objective is addressed through the CDI service ("CDI - Marine data access," n.d.), which provides a meta database to individual datasets (such as samples, timeseries, profiles,

trajectories, etc). In addition, SeaDataNet comprises aggregated datasets, which are regional Ocean Data View ("ODV - SeaDataNet," n.d.) collections of physical measurements from all the European seas ("Aggregated datasets - SeaDataNet," n.d.). Moreover, in addition to maintaining data services at the European level, SeaDataNet established a brokering services with a web-based search interface ("Search portal," n.d.), which allows users to discover marine dataset collections managed by marine data portals worldwide, including the Australia Ocean Data Network (AODN) ("IMOS," n.d.), NOAA National

Centers for Environmental Information ("National Centers for Environmental Information (NCEI)," n.d.) and the World Ocean Database (WOD) ("World Ocean Database | National Centers for Environmental Information (NCEI)," n.d.).

Another important contribution of SeaDataNet to the interoperability of ocean data has been the definition of standards for data, metadata and vocabularies (Buck et al., 2019; Pecci et al., 2020). SeaDataNet provides a searchable vocabulary library ("https://vocab.seadatanet.org/search," n.d.), which is based on vocabulary services based on a NERC Vocabulary Server

("NVS," n.d.) which are technically managed and hosted by the British Oceanographic Data Centre (BODC) ("Common Vocabularies - SeaDataNet," n.d.; Pecci et al., 2020; Schaap and Lowry, 2010). SeaDataNet data products are available in ODV (Ocean Data View) and NetCDF (CF) formats.

### 3.4.4. The Copernicus Marine Environment Monitoring Service (CMEMS)

*Data source infrastructural approach: Distributed marine data management*

Copernicus Marine Environment Monitoring Service (CMEMS) ("Blue markets | CMEMS," n.d.; Le Traon et al., 2019) is the marine component of the EU's Copernicus Earth observation programme ("Copernicus," n.d.). CMEMS provides information on the physical and biogeochemical ocean and sea-ice state for the global ocean and the European regional seas, combining satellite and in situ observations, and numerical models. CMEMS implements a user-driven approach, with user requirements being gathered through user workshops, training sessions, questionnaires and user interactions with the

CMEMS service desk (Le Traon et al., 2019; "User-Driven Approach | CMEMS," n.d.).

CMEMS products and services. The CMEMS catalog includes over 300 standardized quality-controlled products (as of December 2024), of which the most frequently downloaded are real-time global analyses and forecasts, reprocessed and real-time gridded sea-level maps, gridded sea-surface temperature (SST), global ocean reanalysis and Mediterranean Sea regional analyses and forecasts (Le Traon et al., 2019). All CMEMS products (NetCDF format) are freely accessible through a single

internet interface ("Access data | CMEMS," n.d.). The interactive catalog ("Copernicus Marine Data Store | Copernicus Marine Service," n.d.), allows users to select products according to geographical area, parameter, time span, and vertical coverage. Additional capabilities include downloading or visualizing data ("Access data | CMEMS," n.d.), ocean visualization tools ("Visualisation tools | CMEMS," n.d.), product quality dashboard ("Quality | Copernicus Marine," n.d.), ocean monitoring indicators such as health of the ocean, an annual ocean state report ("News | CMEMS," n.d.). Another

interesting capability is the TAC dashboard tracking in situ technology deployed in the ocean ("The Copernicus Marine In Situ Dashboard combines CMEMS & EMODnet data | Copernicus," n.d.).

Interoperability strategy. The CMEMS architecture, which is described in detail by Le Traon et al. (2019), supports the vision of the ocean data ecosystem being a distributed system of systems composed of modular and flexible components. The architecture comprises the following elements: 1. Thematic Assembly Centres (TAC), which gather observational data

from in situ networks and from the Copernicus satellite component. These validated data sets can readily be used for assimilation in models and products ("In Situ Thematic Centre (INS TAC) | CMEMS," n.d.). 2. Monitoring and Forecasting Centers (MFCs), which perform modeling and assimilation. 3. Data and Information Access Services (DIAS), which are five cloud-based platforms that centralize and standardize access to data and products from all Sentinel satellites and Copernicus Services directly from the original sources ("Data and Information Access Services | Copernicus," n.d.). 4. A Central

information system (CIS), which allows searching, viewing and downloading products. 5. CMEMS In Situ Thematic Centre (INS TAC), which collects, processes and quality controls the upstream *in situ* data, such as the ones provided by the Argo

network data from over 3,800 platforms (as of September 2024) collecting vertical physical and biogeochemical profiles worldwide ("Argo," n.d.). ArgoARGO data are delivered in near-real-time, with automatic quality processing, from acquisition to scientifically assessed reprocessed (REP) products, performed with a 24 h framework ("In Situ Thematic Centre (INS TAC) | CMEMS," n.d.). 6. CMES service desk supports user requests.

### 3.4.5. PANGAEA

*Data source infrastructural approach: Distributed marine data management*

PANGAEA is a long-term repository and data publisher for earth and environmental data. Data and metadata in PANGAEA include observational and experimental data, and are freely available via the website ("Data Publisher for Earth & Environmental Science," n.d.; Felden et al., 2023a) and by programmatic access. It currently hosts more than 422,000 datasets, over 26 billion measurements, over 808 national and international projects, and estimated 10,000 datasets published per year (Felden et al., 2023a). Main uses include research data management, long-term data archiving and publication. Data are submitted via a ticketing system, and reviewed for completeness, correctness, quality and interoperability by editorial experts (Felden et al., 2023a). Data and metadata are imported into a relational database for archiving. PANGAEA's key value proposition to the ocean data ecosystem is the high level of quality assurance of data and metadata, and the interoperability-enabling infrastructure. Commitment of the hosting institutions ensures the long-term usability of the archived data. This makes PANGAEA a key player in the ocean data ecosystem, and it is a recommended data repository of numerous international scientific journals and accredited as a World Data Center by the International Council for Science (ICS) (Felden et al., 2023a; "World Data System," n.d.). PANGAEA's main features and strategies for interoperability are as follows:

Data aggregation, search and access, user experience, and citability. PANGAEA's data warehouse (DWH) allows for data aggregation over multiple and sometimes hundreds of studies (spatially and chronologically). Capabilities include calculating daily/monthly/yearly averages and standard deviations. PANGAEA offers several ways to discover and search data (Fig. 7): users can access the PANGAEA search engine on the website, Google Search and Google Dataset Search, and *portals* harvesting PANGAEA metadata (GEO data portal, INSPIRE, DataONE, EMODnet, etc). Pangea offers web services for metadata harvesting and data retrieval and an API ("Interoperability and Services – Data Publisher for Earth & Environmental Science," n.d.). Usability features include an enhanced usability of the website and web-based submission system and rating of datasets via social networks (Felden et al., 2023a). In addition, usage statistics are provided for each dataset ("Data Usage Statistics - PANGAEA Wiki," n.d.). Data references and citations are provided in every export of data, supported by each dataset being associated with a unique Digital Object Identifier (DOI) according to standards of DataCite. Citation can be downloaded in different formats.

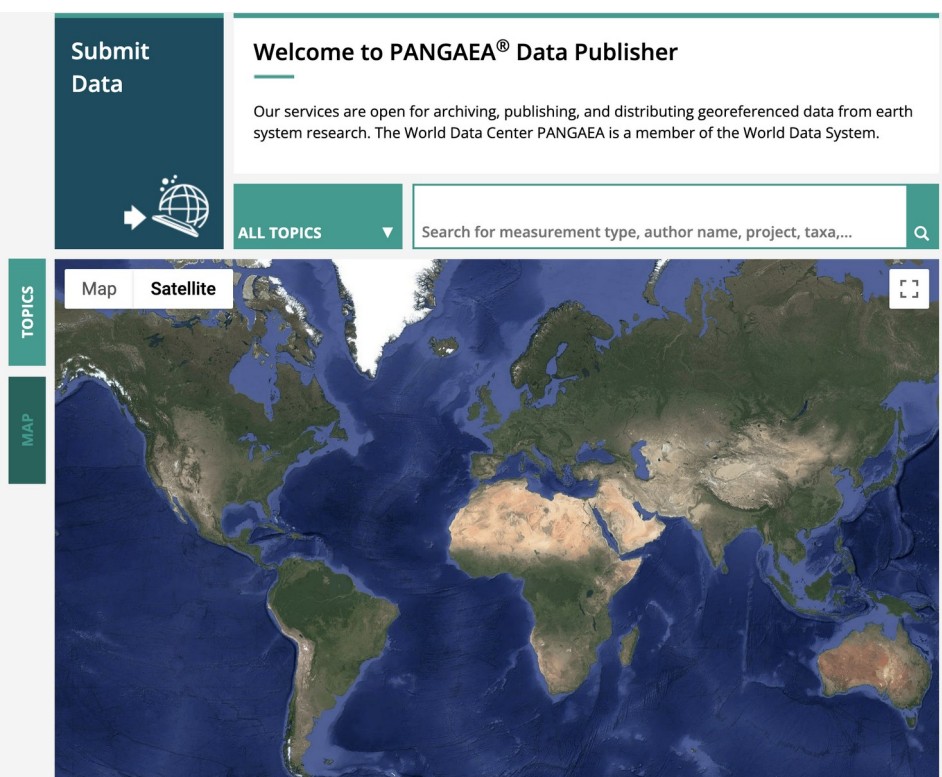

**Figure 7.** A screenshot of The PANGAEA website, demonstrating key features such as search by map, by topic, and by project, and on-demand data submission. (Screenshot from the PANGAEA website (https://www.pangaea.de/), accessed on 26 December 2024. Used under the Creative Commons Attribution License. ("Data Publisher for Earth & Environmental Science," n.d.).

Interoperability strategy. PANGAEA's interoperability is supported by semantic harmonization of the data (comprehensive metadata descriptions, standards, controlled vocabularies and ontologies) and a high degree of structural harmonization (using a relational database). Editors categorize and harmonize the data and metadata, and store it in tables, where rows and columns represent relationships and there are further logical relational connections between different tables. This allows to make the data interoperable, findable and re-usable as independent variables in scientific studies and allows PANGAEA to reach a high level of FAIRness (Felden et al., 2023a). Semantic interoperability is supported by linking each observation with terms from controlled and internationally recognized vocabularies and ontologies. Terminology services include the Integrated Taxonomic Information System (ITIS) ("Integrated Taxonomic Information System," n.d.) and World Register of Marine Species ("WoRMS - World Register of Marine Species," n.d.) taxonomy terminologies, Chemical Entities of Biological Interest ("Chemical Entities of Biological Interest (ChEBI)," n.d.) chemical taxonomy, the QUDT ("QUDT," n.d.) measurements taxonomy, and the Environmental features taxonomy (EnvO) ("PANGAEA Wiki," n.d.; "The Environment Ontology," n.d.).

Further features of interoperability are a terminology catalog (TC), which allows extracting PANGAEA datasets with schema.org/dataset metadata. PANGAEA's interoperability strategy opens the way to dissemination of data and metadata to a large variety of actors in the ocean data ecosystem, including other data sources, search-engine registries, library catalogs and other service providers (Felden et al., 2023a).

### 3.4.6 Rolling Deck to Repository (R2R)

*Data source infrastructural approach: Federated data management*

The Rolling Deck to Repository (Carbotte et al., 2022; "Rolling Deck to Repository (R2R)," n.d.) program is meant to make multidisciplinary routinely acquired shipboard sensor data available for academic research of the marine environment. With over a decade of operations, the R2R program has developed a robust routinized system to transform diverse data contributions from different data providers into a standardized and comprehensive collection of global-scale observations of marine atmosphere, ocean, seafloor and subseafloor properties that is openly available to the ocean science community.

### 3.4.7 DataOne

*Data source infrastructural approach: Federated data management*

DataOne ("Data Observation Network for Earth | DataONE," n.d.; Michener et al., 2011) is a data portal, which provides access to data from multiple member repositories, to support enhanced search and discovery of Earth and environmental data, and to promote best practices in data management. It consists of more than 50 members ("Member repositories | DataONE," n.d.) and over 770,000 datasets (as of December 2024), with usage of more than 17M downloads. DataOnes offerings and services include integrated search across different repositories through a search and discovery platform (https://search.dataone.org/portals), open source tools, API access, metrics visualizations for datasets, service of hosting and maintaining repositories, developing domain-specific ontologies and training, webinars and skills building. DataOne FAIRness assessment is available online shown (Fig. 8) ("DataONE Data Catalog," n.d.). As for December 2024, the online calculated scores were: Findable - 76%, Accessible - 45%, Interoperable - 68%, and Reusable - 51%.

Interoperability strategy. DataONE maintains and develops a family of both general-purpose and domain-specific Web Ontology Language ("OWL - Semantic Web Standards," n.d.) ontologies (Michener et al., 2011), including ProvONE ("The ProvONE Data Model for Scientific Workflow Provenance," n.d.), OBOE - Extensible Observation Ontology ("The Extensible Observation Ontology | NCBO BioPortal," n.d.), DataONE ontology of Carbon Flux measurements for MsTMIP and LTER Use Cases ("The Ecosystem Ontology | NCBO BioPortal," n.d.), MOSAiC ("MOSAiC Ontology," n.d.), Arctic Report Card Ontology ("Arctic Report Card Ontology," n.d.), and Sensitive Data Ontology ("Sensitive Data Ontology (SENSO)," n.d.).

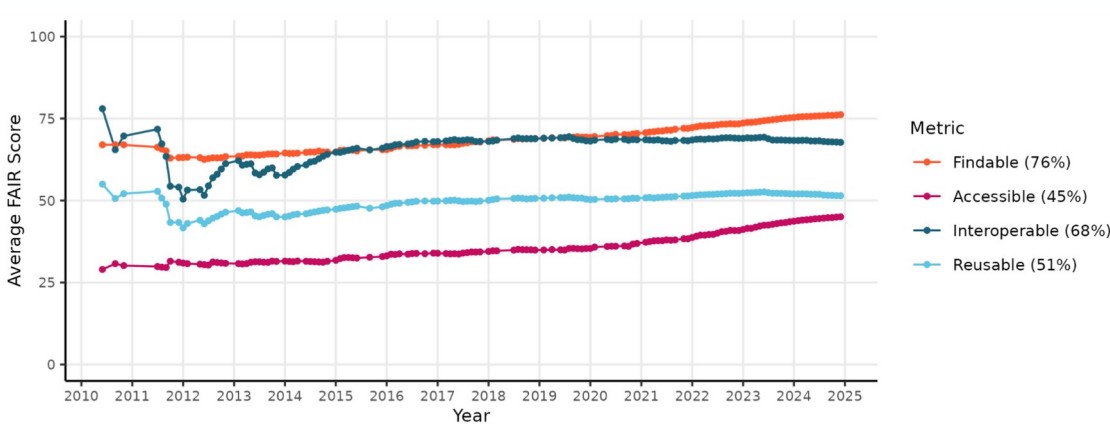

**DataONE FAIR Assessment**

**Figure 8.** The DataOne website provides on-demand FAIRness assessment. (Screenshot from the DataOne website (https://search.dataone.org/profile), accessed on 26 December 2024 ("DataONE Data Catalog," n.d.).*Data source infrastructural approach: Centralized data management Data source infrastructural approach: Distributed marine data management.*

Interoperability strategy. A major objective and challenge in SeaDataNet are to provide an integrated and harmonized access to data resources, using a distributed network approach. This objective is addressed through the CDI service ("CDI - Marine data access," n.d.), which provides a meta database to individual datasets (such as samples, timeseries, profiles, trajectories, etc). In addition, SeaDataNet comprises aggregated datasets, which are regional Ocean Data View ("ODV - SeaDataNet," n.d.) collections of physical measurements from all the European seas ("Aggregated datasets - SeaDataNet," n.d.). Moreover, in addition to maintaining data services at the European level, SeaDataNet established a brokering services with a web-based search interface ("Search portal," n.d.), which allows users to discover marine dataset collections managed by marine data portals worldwide, including the Australia Ocean Data Network (AODN) ("IMOS," n.d.), NOAA National Centers for Environmental Information ("National Centers for Environmental Information (NCEI)," n.d.) and the World Ocean Database (WOD) ("World Ocean Database | National Centers for Environmental Information (NCEI)," n.d.).

Another important contribution of SeaDataNet to the interoperability of ocean data has been the definition of standards for data, metadata and vocabularies (Buck et al., 2019; Pecci et al., 2020). SeaDataNet provides a searchable vocabulary library ("https://vocab.seadatanet.org/search," n.d.), which is based on vocabulary services based on a NERC Vocabulary Server ("NVS," n.d.) which are technically managed and hosted by the British Oceanographic Data Centre (BODC) ("Common Vocabularies - SeaDataNet," n.d.; Pecci et al., 2020; Schaap and Lowry, 2010). SeaDataNet data products are available in ODV (Ocean Data View) and NetCDF (CF) formats.

### 3.4.8. EMODNET

*Data source infrastructural approach: Federated data management*

The European Marine Observation and Data Network ("European Marine Observation and Data Network (EMODnet)," n.d.; Míguez et al., 2019), established in 2009, comprises of more than 150 organizations which gather marine data, metadata, and data products in order to facilitate their accessibility by a broad range of users. EMODnet consists of seven thematic sub-portals, namely bathymetry, geology, physics, chemistry, biology, seabed habitats, and human activities, covering in total over 800,000 datasets. The data are available through open sharing infrastructures such as SeaDataNet ("SeaDataNet - SeaDataNet," n.d.), Copernicus Marine Environment Monitoring Service ("Blue markets | CMEMS," n.d.), European Ocean Biogeographic Information System ("EurOBIS," n.d.), International Council for the Exploration of the Sea ("ICES," n.d.) and the European Geological Data Infrastructure (EGDI) ("EGDI," n.d.), allowing unrestricted access to interoperable European marine data (Tanhua et al., 2019b). EMODNet products and services include the EMODNet map viewer (Fig. 9) ("EMODnet Map Viewer," n.d.), a data products catalog ("EMODnet Product Catalogue," n.d.), EMODNet ERDDAP data server ("ERDDAP," n.d.), and Atlas of the seas ("European Atlas of the Seas | European Marine Observation and Data Network (EMODnet)," n.d.), EMODnet Data Ingestion Portal ("EMODnet Ingestion," n.d.), that was launched in 2017 to further increase the quantity and quality of available European marine data ("EMODnet | Blue-Cloud 2026," n.d.).

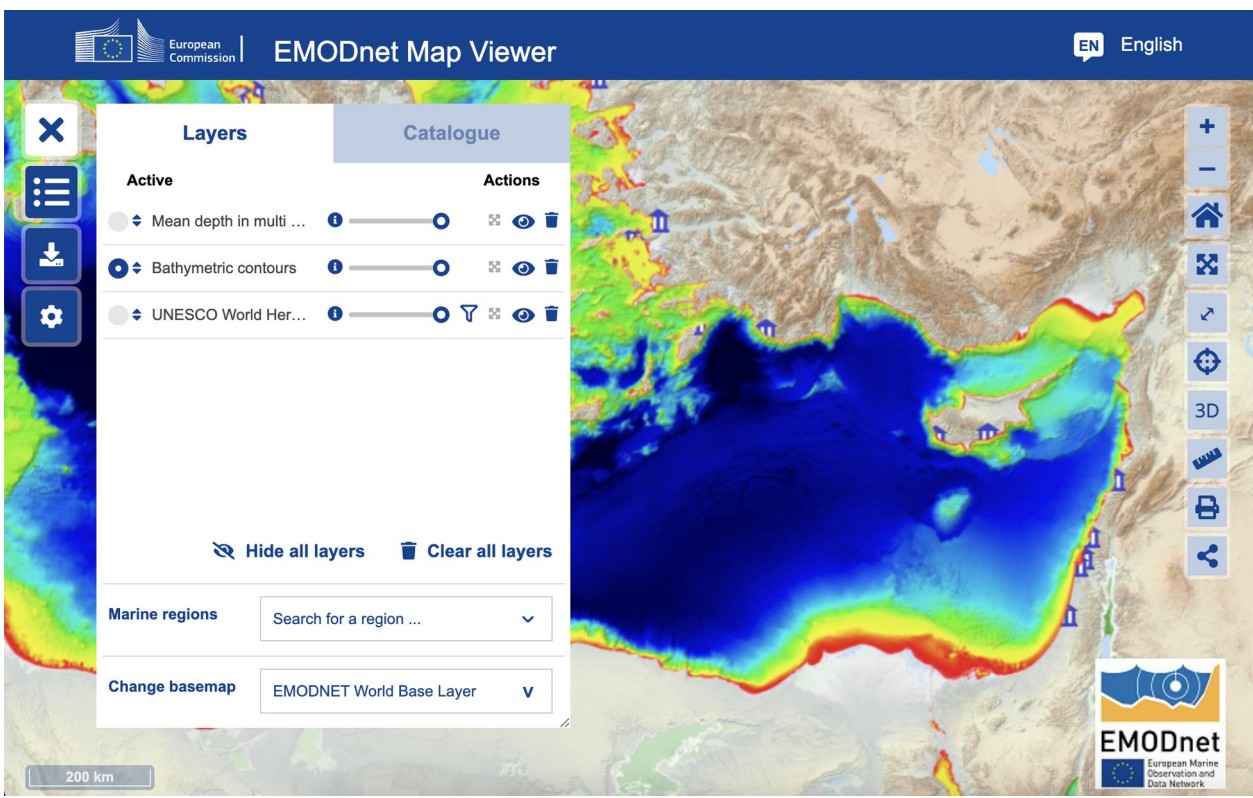

**Figure 9.** EMODnet Map Viewer. (Screenshot from the EMODnet website (https://emodnet.ec.europa.eu/geoviewer/), accessed on 26 December 2024. ("EMODnet Map Viewer," n.d.). Information used in this map viewer was made available by EMODnet ("European Marine Observation and Data Network (EMODnet)," n.d.) founded by the European Commission Directorate-General for Maritime Affairs and Fisheries (EC DG MARE) and funded by the European Maritime Fisheries and Aquaculture Fund (EMFAF).

Interoperability strategy. EMODnet employs a number of strategies for interoperability. For example, EMODnet Biology ("Biology | European Marine Observation and Data Network (EMODnet)," n.d.) aims at implementing and further developing common standards and vocabularies within the ocean data ecosystem (Míguez et al., 2019). EMODnet Chemistry ("Chemistry | European Marine Observation and Data Network (EMODnet)," n.d.) which is a network of more than 100 National Oceanographic Data Centres, has adapted the SeaDataNet services and standards, thus providing easy access to standardized, harmonized and validated marine chemical datasets for all Eu Marine Regions (Míguez et al., 2019).

### 3.4.9 OBIS and OBIS-SEAMAP

*Data source infrastructural approach: Federated data management*

The Ocean Biodiversity Information System (OBIS) ("Ocean Biodiversity Information System," n.d.) is the largest source of information on marine species distribution, providing open-access data from 500 institutions and 56 countries. It encompasses a comprehensive ocean biodiversity data, across species (from bacteria to whales) and habitats (from the ocean

surface to the abyssal and from the tropics to the poles), comprising more than 5,000 datasets with over 119 M records on more than 182,000 species. The datasets are integrated in a way that allows search and mapping by species name, taxonomic level, geographic area, depth, time and environmental parameters (Fig. 10).

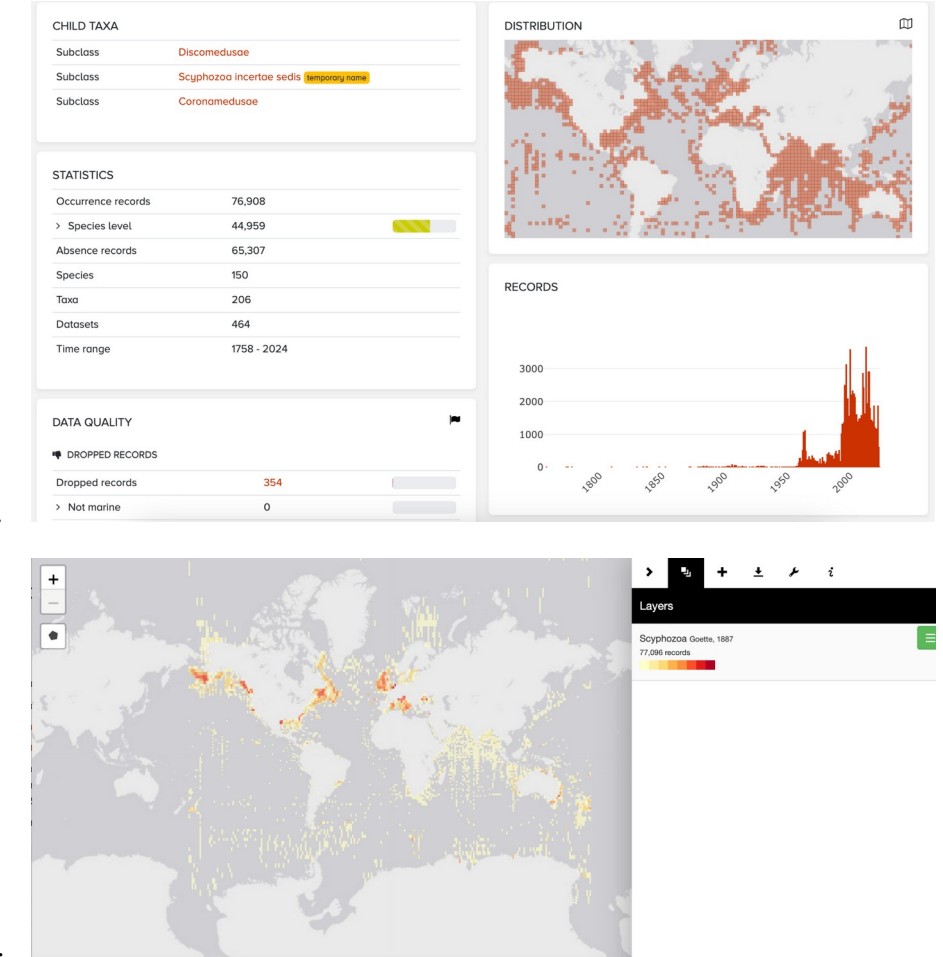

**Figure 10.** A screenshot exemplifying the Using OBIS to view data of the Scyphozoa (Jellyfish, with over 600 datasets) a. using the "Search OBIS" feature (additional information such as records, environmental conditions and top datasets is included on the webpage) and b. Viewing the data using the OBIS Mapper (Screenshots from the OBIS Search website (https://obis.org/?query=jellyfish) and OBIS mapper website (https://mapper.obis.org/), accessed on December 10 2024. Used under the Creative Commons Attribution License ("OBIS-SEAMAP," n.d.).

OBIS consists of a node entitled Ocean Biogeographic Information System Spatial Ecological Analysis of Megavertebrate Populations ("OBIS-SEAMAP," n.d.). It provides a spatially and temporally interactive online database for marine mammal, sea turtle, seabird and ray & shark, and unique applications such as habitat-based density models for marine mammals.

OBIS-SEAMAP statistics, which are available online, show that as of August 2024, OBIS-SEAMAP consists of over 8.3 M records, encompassing over 740 species, over 1580 datasets and 840 contributors.

Interoperability strategy. OBIS relies on a number of external data sources ("Ocean Biodiversity Information System," n.d.). This includes the World Register of Marine Species ("WoRMS - World Register of Marine Species," n.d.) as a taxonomic backbone, Marine Regions ("Marine Regions," n.d.) as a source for geospatial data, and the World Ocean Atlas as a source for information on environmental parameters ("Ocean Climate Laboratory | National Centers for Environmental Information (NCEI)," n.d.).


### 3.4.10 Additional ocean data sources

Additional ocean data sources include the NOAA Environmental Research Division's Data Access Program (ERDDAP) ("ERDDAP," n.d.), The National Centers for Environmental Information ("National Centers for Environmental Information (NCEI)," n.d.), The IOC Ocean and Data Information system Catalogue of Data Sources (ODISCAT) ("IOC Ocean Data and

Information System Catalogue," n.d.; Pinardi et al., 2019), The IOC Ocean Data and Information System (ODIS) ("Ocean Data Information System," n.d.; Pinardi et al., 2019), GEOSS Geoportal ("GEOSS Portal," n.d.), Fishbase (Froese and Pauly, 2022; "Search FishBase," n.d.), The Argo Programm ("Argo," n.d.; Roemmich et al., 2022), European Node of the international Ocean Biodiversity Information System (EuroBIS) ("EurOBIS," n.d.), The Biological and Chemical Oceanography Data Management Office (BCO-DMO) ("Introduction to BCO-DMO | BCO-DMO," n.d.), The US National

Data Buoy Center (NDBC) ("National Data Buoy Center," n.d.), International Council for the Exploration of the Sea ICES ("ICES," n.d.), AtlantOS ("AtlantOS - EuroGOOS," n.d.; deYoung et al., 2019), and the IOOS Environmental Data Server (EDS) ("IOOS Model Viewer," n.d.).

### 3.5. Emerging solutions

We refer to emerging solutions as various efforts made to leverage advanced technologies and methodologies for addressing

the various challenges facing the ocean data ecosystem (Fig. 11). The major approaches taken to address these challenges include 1. Interoperable digital ecosystem. 2. Open-Source Data Platform Tools. 3. Cloud-based Data Management. 4. Unified and curated database portal. 5. Virtual models that simulate ocean conditions using real-time data. 6. AI and ML tools, Ontologies and Semantic Web Technologies. 7. Ocean Data Platform. We now give an overview on key initiatives taking these different approaches.


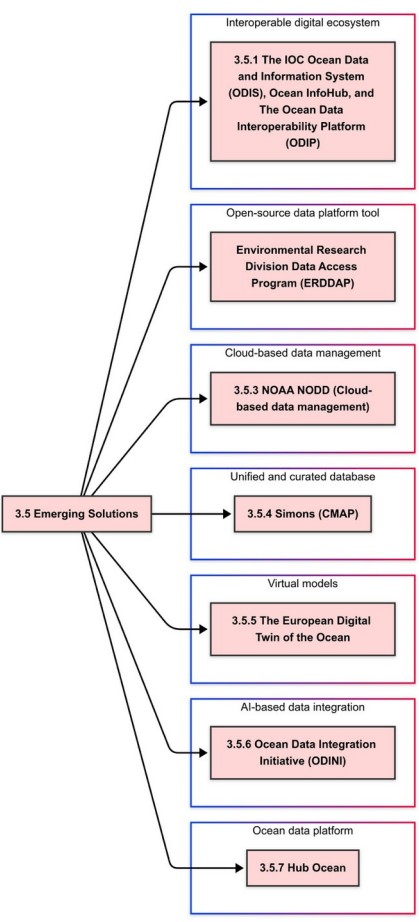

**Figure 11:** A diagram summarizing the examples for emerging solutions discussed in Sec. 3.5, along with brief descriptions of the approach they represent.

### 3.5.1 The IOC Ocean Data and Information System (ODIS), Ocean InfoHub, and The Ocean Data Interoperability Platform (ODIP)


*Approach: Interoperable digital ecosystem*

One of the pioneering efforts to address key challenges in the ocean data ecosystem, is the Global Ocean Data and Information System (ODIS) ("Ocean Data Information System," n.d.; Pinardi et al., 2019)**,** which is an IOC initiative that aims to create a global digital ecosystem that allows for seamless integration and sharing of ocean data and information.

Interoperability strategy. ODIS Interoperability strategy is based on building a partnership of distributed, independent systems voluntarily sharing (meta)data and information (meaning, not a portal or centralized system). ODIS is actively evolving, with its architecture enabling various established and emerging data systems to interconnect. An ODIS Node is a data source that is networked into and part of the ODIS Federation. ODIS harvests (meta)data from all ODIS nodes and builds a collective Knowledge Graph to promote global discovery and action ("Ocean Data Information System," n.d.).

To provide solutions to ocean data challenges, the IOC/IODE also supports the Ocean Data Interoperability Platform (ODIP) ("ODIP," n.d.; Pearlman et al., 2016), which was initiated in 2012 to improve and promote the interoperability of existing marine data management infrastructures. ODIP looks to create an integrated global network by bringing together different regional and national systems. Accordingly, ODIP includes all the major organizations engaged in ocean data management in the EU, US, and Australia.

ODIP addresses the challenge of interoperability through a number of projects, including the following: Interoperability between regional data discovery and access services (ODIP II Prototype 1+). The project includes the SeaDataNet, AODN, USA NCEI regional data portals and interacting with the global IODE-ODP and GEOSS portals, where a brokerage service technological framework is utilized. Another component is the integration of data management for biological and physicochemical marine data (ODIP II Prototype 5), which focused on a use case of marine mammal tracking. Analyze the
usability of the MEOP database ("Marine Mammals Exploring the Oceans Pole to Pole") within the context of the OBIS-ENV-DATA scheme and assess if both data schemes can be matched in order to exchange information between the physical environment and the occurrence of a certain species between both data systems. The ODIP II Prototype 2+ worked on interoperability between the regional cruise summary reporting systems and interacting with the global POGO portal. (US projects "R2R" and "GeoLink", EU project "SeaDataCloud"), the ODIP II Prototype 3+: Sensor Web Enablement (SWE) for
the marine and ocean domain and the ODIP II Prototype 4: 'Cloud-based Virtual Research Environments in the marine domain. Major input for a SeaDataCloud VRE. This VRE focuses on a workflow generating T-S Climatology ("ODIP," n.d.).

### 3.5.2 Environmental Research Division's Data Access Program (ERDDAP)
*Approach: Open-source data platform tool*

The Environmental Research Division's Data Access Program ("ERDDAP," n.d.; "Using ERDDAPTM | National Centers for Environmental Information (NCEI)," n.d.) is an open-source data platform tool, where data are available through interoperable formats, facilitating data interoperability between different data sources in the ocean data ecosystem (Buck et al., 2019; Tanhua et al., 2019b; Vance et al., 2019).


### 3.5.3. NOAA Open Data Dissemination (NODD) Program
*Approach: Cloud-based data management*

NOAA Open Data Dissemination (NODD) Program, ("Cloud Access | National Centers for Environmental Information (NCEI)," n.d.; "NOAA Big Data Program :: North Carolina Institute for Climate Studies," n.d.; "NOAA Open Data
Dissemination (NODD) | National Oceanic and Atmospheric Administration," n.d.), is a partnership between NOAA and

technology companies, to provide open access copies of NOAA's information in the Cloud, to facilitate public use of key environmental datasets (Brett et al., 2020; Buck et al., 2019; Vance et al., 2019). Cloud platforms include Amazon Web Services (AWS), Google Cloud Platform (GCP), and Microsoft Azure. As of August 2024, this program facilitates hundreds of datasets, including NEXRAD Level 2 and 3 radar data, GOES-16/-17 satellite data, National Water Model, Global

Ensemble Forecast System (GEFS), Global Forecast System (GFS) ("List of NOAA Open Data Dissemination Program Datasets | National Oceanic and Atmospheric Administration," n.d.). This approach of publishing NOAA datasets to the cloud has led to increased utilization by users and the reduction of loads on NOAA systems at no extra cost to the government (Brett et al., 2020; Vance et al., 2019). Vance et al. (2019) analyze two specific use cases in detail: GCP's hosting of NOAA's historical climate data from the Global Historical Climatology Network (GHCN) and the transfer of

NOAA's Next Generation Weather Radar (NEXRAD), demonstrating the demand for, and the feasibility of cloud-based access to NOAAs data.

NOAA's NODD addresses the challenge of interoperability by utilizing Cloud platforms for storing, processing, and sharing large volumes of ocean data (Buck et al., 2019). The architecture includes a "data broker", supporting the publishing of NOAA data from federal systems to collaborators' platforms (Vance et al., 2019).


### 3.5.4 Simons Collaborative Marine Atlas Project Simons (CMAP)

*Approach: Unified and curated database portal*

Simons Collaborative Marine Atlas Project Simons (CMAP), (Ashkezari et al., 2021; "Simons Collaborative Marine Atlas Project," n.d.), is a data portal hosting a unified database with manually curated datasets from all sectors of Oceanography,

offering simple interfaces for end-users to retrieve and analyze the data. It involves more than 30 institutions and 440 curated datasets ("CMAP Catalog," n.d.). Simons CMAP datasets include direct observations (e.g.  Argo floats, World Ocean Atlas, Hawaii ocean time series), global multi-decade remote sensing products (e.g., satellite temperature, chlorophyll, altimetry), and global biogeochemical model estimations (e.g. MIT Darwin, Mercator-Pisces), with the aim of facilitating exploration across highly heterogeneous and diverse data.

Key features provided by Simons-CMAP include web-based data visualization, provided by a plotting service and by APIs, which allows for data visualization, analytics, aggregation along time and space axes (e.g., time-series, depth profiles), computing dataset-specific climatology with custom time-frames (e.g., weekly, monthly, quarterly climatologies), and a data catalog ("CMAP Catalog," n.d.), and dataset submission options ("CMAP Data Submission," n.d.), which are available on the website.

The data integration process involves a collection step in which datasets are curated and harmonized according to location and time. Data are annotated with keywords about the data set variables, in order to address the problem of registering variables with different naming conventions, which is a common problem in ocean data harmonization. A web-based

validation tool assists in formatting requirements and identifies errors and outliers during the submission process. A human curation is applied to all data sets, ensuring structure of data and metadata (Ashkezari et al., 2021).


### 3.5.4 The European Digital Twin of the Ocean

*Approach: Virtual models simulating ocean conditions using real-time data*

The European Digital Twin of the Ocean, (Brönner et al., 2023; "European Commission," n.d.; "European Digital Twin Ocean - EDITO," n.d.; Tzachor et al., 2023), is an EU funded project which aims at establishing an interoperable digital representation of the entire global marine and coastal environments by integrating Earth observing, modeling and digital infrastructures. By creating a virtual representation, this initiative aims to provide an environment that c an be used to predict future ocean dynamics.

Data types and product offerings. Data types include satellite data, marine data, advanced models, artificial intelligence, and citizen science (European Digital Twin of the Ocean (European DTO) European Commission (2023), covering physical, chemical, biological, socio-ecological, and economical dimensions. Forecasting periods range from seasons to multi-decades. The intended users are the public, scientists, and policymakers, while the idea is to provide user-driven, interactive and visualization tools that can be applied to topics such as ocean currents and waves, marine life and human activities.

Interoperability strategy. The digital twins aimed at leveraging existing European data infrastructures such as Copernicus Marine Service (CMEMS), Copernicus Data and Information access services (DIAS) ("Data and Information Access Services | Copernicus," n.d.), which is a digital infrastructure that provides access to Sentinel data and Copernicus information products, and European Marine Observation and Data Network (EMODnet), into a single digital framework, providing a platform for users to easily access marine data and derive insights ("European Digital Twin Ocean - EDITO," n.d.).

Infrastructural elements and related European projects. EDITO ("European Digital Twin Ocean - EDITO," n.d.) is the core infrastructure of the EU DTO (developed by Mercator Ocean International and the Flanders Marine Institute) ("Mercator Ocean - Ocean Forecasters," n.d.; "Vlaams Instituut voor de Zee," n.d.). The first prototype is open and accessible ("EU DTO Platform," n.d.), and allows to explore the data in time and space, or to use the data and tools for creating predictions for the impact of climate change and human activity. Other related European research projects include the EDITO-Model Lab, which develops ocean models for the European DTO. The Iliad Digital Twins of the Ocean ("Digital Twins of The Ocean - The Iliad Project," n.d.), funded under the Green Deal Call which aims to establish an interoperable, data-intensive, and cost-effective Digital Twin of the Ocean (DTO) (Parkinson et al., 2024), with currently with 56 international partners and with over 300 data products as of October 2024. Immerse ("IMMERSE project website," n.d.), which develops numerical high resolution ocean circulation models. Blue-Cloud 2026 ("Blue-Cloud 2026," n.d.) and AquaINFRA ("AquaINFRA," n.d.), connecting data on the marine and coastal environment, biodiversity, and the water cycle with the

 'Blue Economy, by bringing together leading European marine data infrastructures and networks, including SeaDataNet, EurOBIS, Euro-Argo, ICOS, SOCAT, ENA, EMODnet, and CMEMS ("European Digital Twin of the Ocean (European DTO) - European Commission," n.d.).

### 3.5.5 The Ocean Data Integration Initiative (ODINI)

 *Approach: AI-based data integration*

The Ocean Data Integration Initiative ("Discover – ODINI," n.d.; Sagi et al., 2020), is an academic research project aimed to facilitate the utilization of the large amount of available data, which currently relies on time and labor-intensive manual execution of the data integration process (Sagi et al., 2020). ODINI's approach is to automate the ocean data integration process through development and implementation of AI ontology-based data integration tools.

 The ODINI platform. ODINI's platform allows users to semi-automatically integrate data from a wide variety of data sources, by addressing the three phases of the ocean data integration process: *discover*, *merge*, and *evaluate* ("Discover – ODINI," n.d.; Sagi et al., 2020). In the *Discovery* phase, the list of possible candidate datasets for the project is compiled. In the *Merge* phase, candidate datasets are harmonized semantically, computationally, and geographically to form one large and coherent dataset. In the *Evaluate* phase, the results are analyzed to assess quality, coverage, and bias, and appropriate
 corrections are made to support assertions made over the data. As of December 2024, ODINI is available for researchers to use for the discovery and merging of datasets.

Interoperability strategy. ODINI's unique contribution is in allowing any datasets to be integrated over any set of concepts in the oceanographic domain. ODINI maintains a large integrated ontology constructed from several of the fields ontologies such as ENVO and SWEET. The ontology is being developed by evaluating existing ontologies for domain fit and
 correctness (Zaitoun et al., 2023) and constructing new ontological fragments from sets of scientific papers of domain sub fields. In order to generate these fragments, custom AI models are being trained using a unique verbalization method to generate textual fragments from ontological sub-trees (Zaitoun et al., 2024).

Infrastructural elements. The system comprises a set of cloud-based micro-services. The discovery services allow users to upload their own datasets or mass-download datasets from external repositories through the DataONE," n.d.) data portal. The
 link service disambiguates duplicate records and overlapping datasets using a unique generalized entity resolution approach ("Generalizing Spatio-Temporal Entity Resolution / Qais Abou Housien ; supervised by Tomer Sagi - Haifa University," n.d.). After the dataset collection is finalized, the user selects a mediated schema - a set of measurement types from the ODINI ontology that they wish to integrate the collected datasets on. The schema matching service is then invoked to match the datasets collected into the mediated schema. These steps are followed by a user evaluation procedure using a mapping
 evaluation service based on the VOWLMap tool (Guerreiro et al., 2021) to verify and amend the matches. Finally, the

datasets are unified into a standard CSV structured file where every row represents a single measurement type in space and time.

### 3.5.6. Hub Ocean

*Approach: Ocean data platform and product offering*

Hub Ocean ("HUB Ocean | Unlocking Ocean Data," n.d.) is an independent non-profit foundation, which is developing an ocean data platform as well as data products to support new approaches to ocean governance. The platform aggregate ocean data from various sources, allowing users to access, visualize, and analyze data from a wide range of sources in a single cloud-based environment and by API. Additional offerings include access to data bundles, which are thematic groups of datasets packaged together, and access to cloud-based data science workspaces and visualization.

**Interoperability strategy.** The Hub Ocean Ocean Data Platform addresses the challenge of interoperability by gathering, fusing and analyzing data from diverse sources and will continue development to expose the data catalog and the data through different common standards and formats. The data catalog includes data from large open-source datasets (e.g. World Ocean Data Analysis Project – GLODAP) ("Global Ocean Data Analysis Project (GLODAP) - Global Ocean Monitoring and Observing," n.d.). Derivative data in the form of the prototype Ocean Sensitive Areas will soon be available. Unique (industrial) datasets are also available (e.g. acoustic krill fishing data, WWF Ocean Futures and Norwegian Salmon Parasites data "Lusedata".) ("The Ocean Data Platform - HUB Ocean," n.d.). Users can access the various datasets via API and cloud-based workspaces.

### 4. Roles in the ocean data ecosystem

A data ecosystem can be considered as consisting of four main concepts: actors and roles, relationships and resources (Oliveira et al., 2019). In the previous sections we described the main actors comprising the marine data ecosystems. We now give a short overview of the roles played by these different actors.

- **Data users.** As discussed above in section 3.1, in the scope of this paper, the main users in the ocean data ecosystem are marine researchers. Other users in the data ecosystem may be software developers, marine device technology developers, professionals in the marine industry, government, regulation and funding agencies, and the general public.

- **Data producers**. Data producers are the actors that are the root source of the data. On the most basic levels these are researchers and technicians performing various tasks of data collection and sharing. These can be part of academic institutions, as well as regional, national and international ocean observing organizations and frameworks,

such as The Global Ocean Observing System (GOOS), European Ocean Observing System Framework (EOOS), the US Integrated Ocean Observing System (IOOS), the Joint Technical Commission of Oceanography and Marine Meteorology in situ Observing Platform Support (JCOMMOPS), and many others.

- **Data providers.** Data providers are responsible for linking data producers and data users, allowing the latter to utilize the wealth of ocean data collected worldwide. Data providers may be aggregators or storers (e.g. database/ portal/emerging data integration solutions), or service providers of associated product offering and services. In our review of data sources and their product offering and emerging solutions, we covered different types of solutions, strategies, workflows and approaches for data aggregation, data integration, harmonization and storing. Examples of data sources include: PANGAEA, CMEMS, EMODnet, SeaDataNet, WOD, DataOne, OBIS and OBIS-SEAMAP, Rolling Deck to Repository (R2R), and WoRMS. Examples of emerging solutions we covered include The IOC Ocean Data and Information System (ODIS), Ocean InfoHub, and The Ocean Data Interoperability Platform (ODIP), Environmental Research Division's Data Access Program (ERDDAP), NOAA Open Data Dissemination (NODD) Program, Simons Collaborative Marine Atlas Project Simons (CMAP), The European Digital Twin of the Ocean (EDITO), The Ocean Data Integration Initiative (ODINI), and Hub Ocean.

- **Drivers.** The term drivers refer to the driving forces standing behind the continuous development of the ocean data ecosystem. These set the goals and directions, define key challenges and recommendations, provide the funding, and drive collaborations, eventually driving for the development of innovative solutions. In our data ecosystem model, an important driver of the marine data ecosystem is the society element, through guiding principles such as FAIR, key initiatives such as the UN Ocean Decade, International Oceanographic Data and Information Exchange program (IODE), and the Framework for Ocean Observing (FOO). Policies such as The General Data Protection Regulation (GDPR), The IOC Oceanographic Data Exchange Policy, EuroGOOS Data Policy, The Marine Strategy Framework Directive (MSFD), The Infrastructure for Spatial Information in Europe Directive (INSPIRE), as well as targets such as UN Sustainable Development Goals and the AICHI targets also serve as a driving force. Other important drivers are associated with the Interoperability tools and framework element, through standards and best practices, such as the Ocean Best Practices, and the Essential Ocean Variables (EOV), as well as stakeholders, such as SeaDataNet, which plays a key role in developing standardization, and serve as a driving force within the ecosystem.

## 5. Summary and conclusions

Over the past two decades, ocean science has undergone a profound transformation in the availability, accessibility, and management of data. While in the early 2000s, oceanographic data were largely confined within research institutions, with limited standardization and few mechanisms for data sharing, the development of data repositories and interoperability tools,

including platforms like SeaDataNet, PANGAEA, DataOne, and EMODnet, has enhanced data accessibility substantially. Despite significant improvements in accessibility and availability, ocean data remains highly fragmented. Researchers often face considerable challenges in finding, accessing, and integrating the datasets they need. The lack of standardized data management structures and inconsistent metadata protocols (Brett et al., 2020; Tanhua et al., 2019b) further complicates data sharing and synthesis, making it difficult to conduct seamless, cross-disciplinary, and geographically broad analyses.

Our analysis highlights several key trends that will shape future evolution of the ocean data ecosystem, serving as both guiding principles and essential requirements for continued development. These trends reflect emerging technological advancements, evolving research needs, and increasing demands for seamless data integration. Below, we discuss the most significant factors expected to drive progress in ocean science data.

## 5.1 FAIR ocean data and data democratization

A most fundamental requirement for any ocean data solution is that it satisfies the need for the data to be Findable, Accessible, Interoperable, and Reusable, as defined by the FAIR principles (Tanhua et al., 2019b; Wilkinson et al., 2016). This is critically important for driving the ocean data market towards data democratization, that is, data generated and funded by various national and international government programs, should be freely and easily available to the public (Buck et al., 2019). We note however that while ocean data literature strongly promotes more open and democratic access to data, ocean scientists, who are responsible for the collection of data, often fail in sharing it, unintentionally taking a somewhat contrasting approach. To account for this discrepancy, which is common in various scientific disciplines (Borgman, 2017), efforts should be made to enhance active data sharing, by facilitating the process of data upload to open access repositories on one hand, and by crediting scientists who do so on the other.

## 5.2 Comprehensive product offering and needs-based solutions

Data sources are moving towards offering comprehensive product offerings such as data viewers, maps and geospatial products, climatologies, and atlas. This is well exemplified by CMEMS, which provides 275 standardized quality-controlled products of satellite and in situ data. Moreover, it is acknowledged that solutions in the ocean data ecosystem should be prioritized and designed based on user needs (Ashkezari et al., 2021; Buck et al., 2019; Carbotte et al., 2022; Eschenbach, 2017; Tanhua et al., 2019b). The UN Ocean Decade Data & Information Strategy (Intergovernmental Oceanographic Commission, 2023) includes the requirement for science projects to be evaluated for their fitness "for specific purposes and needs and their ability to deliver insights that are urgently needed to enhance decision making at all levels". Developing needs-based solutions, or "fit-for-purpose" products in the ocean data ecosystem, requires answering the needs and

requirements set by all actors in the ocean observing value chain, including the data users and the data source stakeholders. It also calls for involving users from the initial stages of project definitions. Moreover, it is important to address the EOVs requirements, which have been defined by the ocean observing community (Tanhua et al., 2019b).

A useful tool for identification of user-needs for designing solutions in the world of ocean data is the conduction of surveys and interviews of researchers, which allow identifying user needs and challenges within the diverse disciplines of oceanography (Ashkezari et al., 2021; Carbotte et al., 2022; Lima et al., 2022). Other examples include The Rolling Deck to Repository program (R2R, Carbotte et al., 2022), which partners with the science user community, ship-operators, and the NCEI archive.

## 5.3 The infrastructural strategies of the ocean data ecosystem

The ocean data ecosystem is moving towards a "system-of-systems" approach (Buck et al., 2019; Carbotte et al., 2022; Nativi et al., 2021). New technology architectures and data processing workflows are constantly evolving, allowing aggregation of data from various regions and scientific disciplines. This supports the transition from "portal model", where users download data from repositories, to a "service model" where the user can find new ways to interact and create value from the data (Buck et al., 2019).

We identify three key structures of data ecosystem architectures, namely *centralized*, *distributed*, and *federated*. Examples for centralized data ecosystem approaches can be found in long term time series datasets as the Bermuda Atlantic Time-series Study ("BATS | BIOS," n.d.; Steinberg et al., 2001) and Hawaii Ocean Time-series ("HOT : the Hawaii Ocean Time-series," n.d.; Karl and Lukas, 1996). In a *distributed network*, data management may be centralized, aggregating data from various external datasets. In this architecture the network may encompass multiple geographical locations, which requires coordination between the different participants (Ramalli and Pernici, 2023). We have reviewed such major distributed network frameworks in the ocean data ecosystem, including the CMEMS ("Blue markets | CMEMS," n.d.; Le Traon et al., 2019), SeaDataNet (Pecci et al., 2020; Schaap and Lowry, 2010; "SeaDataNet - SeaDataNet," n.d.) and DataOne ("Data Observation Network for Earth | DataONE," n.d.; Michener et al., 2011). A *federated network* allows multiple and possibly geographically distributed networks to work together (Ramalli and Pernici, 2023). It is generally accepted in the field of ocean data that such federated data networks may facilitate the sharing of data and connecting disparate ocean databases (Brett et al., 2020; Tzachor et al., 2023). EMODnet ("European Marine Observation and Data Network (EMODnet)," n.d.; Míguez et al., 2019) is an example of a federated infrastructure. Another infrastructural strategy emerging in the ocean data ecosystem is cloud-based data access. This approach can be exemplified by NOAA's Open Data Dissemination (NODD) Program ("Cloud Access | National Centers for Environmental Information (NCEI)," n.d.), where NOAA's data are integrated into cloud-based tools (Vance et al., 2019).

**5.4 Interoperability tools**

We reviewed a number of interoperability tools such as standards, vocabularies and marine ontologies. These tools continue to be developed and are recognized as key enablers for data interoperability of a future data ecosystem. Actors within the data ecosystem (e.g. SeaDataNet, EMODNet and others) have been involved in the development of standards and vocabularies and aim at implementing and further developing interoperability tools (Míguez et al., 2019; Pecci et al., 2020; Schaap and Lowry, 2010). Advances in searchability such as schema.org have the potential to improve the access to data (Buck et al., 2019; Tanhua et al., 2019b). Since finding and navigating ontologies remains challenging, actors within the ecosystem continue to contribute to the development of ontologies, as well as other supporting tools such as automating annotations through monitored machine learning ("International Metadata Standards and Enterprise Data Quality Metadata Systems | DataONE," n.d.).

**5.5 Automation of the data workflows and AI advances**

With the rapidly growing amount of ocean data, the automation of the data workflows while maintaining quality is a major market need (Tanhua et al., 2019b). For example, The World Ocean Database and PANGAEA are long-term repositories, utilizing manual expert dependent workflows. These workflows are limited by the time and labor associated with data submissions and handling, which will increase with the growing volume and complexity of data (Felden et al., 2023a). New tools and workflows are required, and being developed, to help with the challenge of maintaining data quality, while integrating large and diverse datasets. As for other data ecosystems (Curry et al., 2021) artificial Intelligence (AI) technologies play a role in enhancing ocean data integration, by automating the processes of data discovery, merging, and evaluation. A review of implementation of AI tools in marine sciences is provided by Song et al., (2023). As oceanographic research generates vast amounts of diverse data, AI tools can facilitate the integration of datasets that lack common schemas and were collected using different methodologies (Sagi et al., 2020). Soares et al. (2018) utilized Semantic Web standards to process heterogeneous data streams for real-time event detection and improved knowledge interoperability in maritime environments. Danyaro et al. (2022) review the use of Machine learning (ML) to enhance the interoperability of metocean data (the combined effect of meteorology and oceanography), allowing for more efficient monitoring and automation in industries reliant on ocean data, such as oil and gas. AI technologies can also enhance ocean data integration by processing multiple datasets, identifying ships, and tracking movements, aiming to improve maritime safety, security, and environmental protection through open-data analysis (Mdakane et al., 2023).

**5.6 Archiving of historical databases and virtual research environments**

Additional noticeable trends include archiving of historical databases such as the NCEI World Ocean Database (Levitus et al., 2013; "World Ocean Database | National Centers for Environmental Information (NCEI)," n.d.), which is aimed at increasing the amount of data availability to the scientific community, focusing on specific identified needs such as high-

resolution CTD data and additional historical chlorophyll, nutrient, oxygen, and plankton data. Another noticeable trend is that of virtual research environments, such as the SeaDataNet virtual research environment ("VRE - SeaDataNet," n.d.), which are aimed to provide software to interpolate, analyze and visualize marine observations. Such systems will be accessible online using remote computing power and provide virtual workspaces for online collaboration.

Looking ahead over the next two decades the ocean data ecosystem is set to undergo further transformations, driven primarily by dramatic growth in the amount and diversity of oceanic data, and by rapid technological developments. The expected increase in data availability and diversity is a natural continuation of the growing use of autonomous and remote sensing platforms, expansion of global observation networks, and improved ability to collect and analyze new data types such as environmental DNA and underwater imagery. Advances in data collection methods results in an unprecedented influx of ocean data each day, often in real-time, propelling ocean research into the era of big data that is characterized by vast volumes, diverse formats, and widely dispersed datasets (Tanhua et al., 2019).

The synthesis presented in this review points to a decisive transition: the ocean data ecosystem is evolving from a patchwork of independent repositories into a globally connected, service-oriented network. Interoperable standards, shared vocabularies, and federated cloud infrastructures are dismantling the old "portal-download" paradigm and enabling machine-actionable data flows across disciplines and borders. The next decade will likely see near-real-time discovery, access, and fusion of ocean observations - from autonomous sensors to satellite archives - within a seamless digital environment.

A defining feature of this emerging landscape is the integration of advanced artificial intelligence (AI) and machine-learning methods. As the volume, velocity, and variety of ocean data continue to grow, AI is becoming indispensable for automated quality control, feature extraction, and pattern recognition across heterogeneous data streams. Deep-learning models can already detect mesoscale eddies, track marine heatwaves, and identify biodiversity "hot spots" in vast image libraries. Looking ahead, AI-driven digital twins of the ocean will couple observation networks with predictive models to deliver near-instant forecasting and scenario testing, transforming both basic research and operational decision-making.

This technological leap will also reshape the social and governance dimensions of the ecosystem. Trust, transparency, and inclusivity, embodied in the FAIR, CARE, and TRUST principles, remain critical as AI systems begin to make or recommend management choices. International initiatives such as the UN Ocean Decade, the Ocean Data Action Coalition, and IOC/IODE programs are fostering open access and shared stewardship, while highlighting the need for clear policies on data provenance, algorithmic accountability, and equitable participation. Ethical AI frameworks, together with persistent identifiers and accreditation schemes, will help maintain confidence in automated analyses and encourage collaboration across nations and institutions.

Ultimately, the ocean data ecosystem is poised to become an active, intelligent engine for discovery and policy. By linking high-resolution observations, interoperable standards, and AI-powered analytics, it will enable rapid synthesis of knowledge for climate adaptation, biodiversity conservation, sustainable fisheries, and the broader blue economy. In this envisioned

future, ocean data are not merely archived, they are continuously analyzed, interpreted, and applied, allowing scientists, governments, and society to anticipate and respond to a changing ocean with unprecedented speed and precision.

In summary, by mapping the market landscape in the field of ocean data, this review paper is meant to enable  the reader, especially the new entrant to the ocean data field, to establish an understanding of the ocean data sector. To maintain long-term relevance, the ecosystem model presented is aimed to be used as a tool in further characterization of the ocean data ecosystem. The examples given are not exhaustive, and the reader may further identify relevant examples within their domain. The model has been placed as an open online resource, describing the elements of the data ecosystem as concepts, and examples as instances with relationships. The model is open to be further validated and refined by the ocean data community. The results bridge gaps between different disciplines and levels of familiarity with ocean data. We provide an up-to-date analysis of ocean data sources and emerging solutions and a summary of relevant data standardization efforts such as marine standards, vocabularies, and ontologies. By characterizing the ocean data ecosystem, we intend to assist the scientific community in identifying the gaps, current needs and future vision of the ocean data ecosystem. This work aims to contribute to the development of needs-based solutions, components, products, services, and technologies, thus contributing to the evolution of the ocean data ecosystem and promoting data-based ocean research.

**Author contribution**

MBH developed the model framework and led the writing of the manuscript with contributions from YL and TS.

**Competing interests**

The authors declare that they have no conflict of interest.

**Acknowledgments**

This work was partially supported by the Data Science Research Center (DSRC) at the University of Haifa through the Israel PBC grant *Advancing Data Science to Serve Humanity and Protect the Global Environment* [grant no. 100009443]

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
