# Peer review of "An overview of the ocean data ecosystem"

_EGUsphere, 2025_

## Author Response (AR1)

We thank the reviewers for their constructive comments and suggestions. We have addressed all the points raised and implemented the suggested changes. Main revisions include:

- Deleting/replacing unnecessary/unclear figures and adding a number of new ones
- Addition of interactive online resource that will be used as a long-term reference after the paper after it is published, ensuring it continues to be up-to-date (*https://odini.net/OceanDataEcoSystem/, with an interactive visual representation available at https://webvowl.odini.net*)
- Revision of  the text to ensure its clarity, readability, accuracy and consistency ..

We hope that the manuscript is now ready for publication in Ocean Science.

Our detailed point-by-point response to the reviewers' comments and suggestions is found below. All page/line/reference/figure numbers refer to the clean version of the revised manuscript. Reviewers' comments are in regular text, **our responses are in bold** and *new text from the manuscript is in bold italics*.

**Reviewer 1**

Scientific significance

The manuscript is a detailed and predominantly balanced review of published data management practices and infrastructure that summarises the two decades of progress made in the community, the paper is a succinct summary of outputs shared within communities including OceanOBS'19 decadal conference outputs, AGU/EGU informatics, the research data alliance, the European IMDIS community, and likely more that I am not aware of. The manuscript maps these outputs on to the data ecosystem concept of Oliveira et al. (2019). Such a review is a valuable and useful contribution to the literature, albeit it may fall out of date relatively quickly with the pace of advancements in the environmental informatics.

**Answer: we thank the reviewer for the positive feedback. In order to ensure the paper relevancy, we added a web-based resource (https://odini.net/OceanDataEcoSystem/) that will be updated regularly, maintaining the paper up-to-date after its publication. The additional tool is referred in the text, as follows (lines 184-188):**

**"*For the purpose of providing an interactive map, we created an online ocean data ecosystem ontology available at https://odini.net/OceanDataEcoSystem/, with an interactive visual representation available at https://webvowl.odini.net. This map is a long-term reference that may be updated and extended. To facilitate contributions and comments from the public, we make the ontology publicly available on GitLab (https://gitlab.com/odini_dev/data-ecosystem-ontology) and invite readers to suggest additions and corrections (ODINI / Data Ecosystem Ontology · GitLab, n.d.; Ontology Documentation generated by WIDOCO, n.d.; WebVOWL, n.d.).*"**

The manuscript has been submitted to a journal special issue with the scope "reviews and perspectives" papers, looking back at how ocean sciences have advanced over the last 20 years and looking forward to how they might advance over the next 20 years.

The manuscript addresses significant data management advances made in the last 20 years but there is less emphasis on the next 20 years. The oceanography and informatics community faces significant challenges over the next decades including (not an exhaustive list); the increasing volume of data (it is not uncommon to collect petabytes or more of data during a single expedition), the types of data (recent advances include imagery, acoustics, genomic data), move towards more real time data flows supporting increasingly complex digital infrastructure such as digital twins and AI, and the challenges in data citation and acknowledgement of data usage when it is shared.

The manuscript presents many of these as trends in its final section but does not bring them together with a vision or summary on how the data ecosystem might advance over the next 20 years. Such an addition to the end of the manuscript would fully align the paper to special issue scope and bring what is an extensive and detailed review to a succinct conclusion for the reader.

**Answer: We thank the reviewer for this constructive comment. To put more emphasis on the expected evolution of the ocean data ecosystem in the coming 20 years, the following text is now included in section 5 (lines 971-979):**

**"*Looking ahead over the next two decades the ocean data ecosystem is set to undergo further transformations, driven primarily by dramatic growth in the amount and diversity of oceanic data, and by rapid technological developments. The expected increase in data availability and diversity is a natural continuation of the growing use of autonomous and remote sensing platforms, expansion of global observation networks, and improved ability to collect and analyze new data types such as environmental DNA and underwater imagery. Advances in data collection methods results in an unprecedented influx of ocean data each day, often in real-time, propelling ocean research into the era of big data—characterized by vast volumes, diverse formats, and widely dispersed datasets (Tanhua et al., 2019). As in other research fields, the fundamental changes in the characteristics of available ocean data, together with dramatic developments in AI technologies opens the way to data-driven research directions, as exemplified by the digital twin of the ocean initiatives.*"**

Scientific quality

The manuscript appropriately references recent literature extensively to support the arguments made by the authors. I agree that the community consensus in published literature is toward more open and democratic access to data. However, this does not reflect the consensus of the entire ocean community with significant differences in data culture present across oceanography. These are well covered in "big data, little data, no data" by Christine I. Borgman. Acknowledging the different data cultures which begin at the definition of what data are would add value to the manuscript.

**Answer: We thank the reviewer for pointing to this discrepancy. In the revised manuscript we address it by adding the following text to 5.1 (lines 989-994):**

**"*We note however that while ocean data literature strongly promotes more open and democratic access to data, ocean scientists, who are responsible for the collection of data, may often be apprehensive, lacking the incentives or resources for sharing the data, and thus taking a somewhat contrasting approach. To account for this discrepancy, which is common in various scientific disciplines (Borgman, 2017), efforts should be made to enhance active data sharing, by facilitating the process of data upload to open access repositories on one hand, and by crediting scientists who do so on the other.*"**

The NOAA big data program now goes by another name "NOAA Open Data Dissemination (NODD) Program" in its most recent iteration and this section may be in need of update, more information at https://www.noaa.gov/information-technology/open-data-dissemination .

**Answer: We reviewed the section to verify its correctness with the recent iteration of the NODD Program and fixed the name of the program**

Presentation quality

The manuscript is well written with appropriate use of English language. It is logically organised using the data ecosystem concept to structure the review makes what is a very detailed review accessible to a broad audience. Figures and tables are appropriate to the manuscript.

**Answer: we thank the reviewer for the positive feedback. We note that following comments from reviewer #2 we have made substantial changes in the figures, which contribute substantially to the paper's clarity.**

There is an inconsistency in the use of the major acronyms used throughout the manuscript notable examples include ARGO (historical term) vs. Argo (Argo is the current term) and netCDF vs NetCDF (NetCDF is correct I believe).

**Answer: Major acronyms have been corrected for consistency throughout the text. Importantly, the term "ARGO" was replaced with "Argo" and the term "netCDF" was replaced with "NetCDF".**

Reviewer 2

The manuscript titled "An overview of the ocean data ecosystem" provides a large review of the ocean data ecosystem with a detailed, but not exhaustive, list of definitions, data sources and data product offerings. While the information presented is factually correct, it is difficult to understand and follow. The main goal of the paper was to produce an easy to navigate map, which was not presented. Whilst this paper is a valuable contribution to the literature, it requires major revisions to be a useful resource to the community and structured such that it wouldn't quickly go out of date.

**Answer: We thank the reviewer for their thoughtful feedback and for recognizing the value of our manuscript as a contribution to the literature. We appreciate the comments regarding the clarity and structure of the paper, and we have revised the manuscript with the aim of improving its understandability, readability, and overall usefulness to the community. Importantly, we made changes in the figures - deleting/replacing unnecessary/unclear ones and adding a number of new ones; created an interactive online ocean data ecosystem ontology that will be used as a long-term up-to-date reference after the paper after it is published; and modified the text to ensure its clarity.**

**In addition, in order to put in context the various changes that have been made throughout the manuscript, the following text was added to the manuscript (lines 177-182): "*By developing a structured conceptual model of the ocean data ecosystem and providing illustrative examples, we intend to support readers in navigating this complex and evolving space. Rather than presenting an exhaustive and potentially quickly outdated inventory, the model is intended to help readers identify and characterize relevant examples (e.g. stakeholders, societal elements, integration tools, data sources and emerging solutions) that are most applicable to their specific domain, use case, or geographic context.  By presenting a flexible and structured framework, the model will serve as a tool to enable to add new developments and technologies, as they emerge in the ocean data ecosystem.*"**

**We hope that by presenting a flexible and structured framework, the paper will serve as a long-term reference that remains relevant as new developments and technologies emerge in the ocean data ecosystem.**

General Comments:

The main goal of the paper was to produce an easy to navigate map but no such map is presented in the paper. Two schematics are presented, Figure 2 and 3, however relationships between elements examples is missing. The following description of each element and examples for the ecosystem is presented through a long, but not exhaustive, list that becomes hard to follow and understand. Therefore, the main goal of the paper to have an easy to navigate map is lost. A figure summarising everything would go a long way to helping readers understand and be useful as a resource in the future. A spaghetti diagram/map, so that shows the various divisions of actors/stakeholders and how their roles are interconnected to each other and the resources to better demonstrate the relationships amongst them all. For example, a data aggregator would interact with an actor that provides data, as well as end-users, all while using the resources of data, software and infrastructure to deliver the aggregation. If this was a digital map/resource, it could be updated into the future as the ecosystem continues to develop and grow.

**Answer: We thank the reviewer for this valuable and constructive comment, which has been addressed in the manuscript as follows:**

- **We deleted the original Fig. 1**
- **We replaced Fig. 2 with a new schematic of the ocean data ecosystem and its elements (now figure 1). The new figure denotes the sections and subsections in which the different elements are discussed, thus serving as a high level overview of the ocean data ecosystem model, and the way it is presented in this paper.**
- **For each of the sections 3.2 - 3.5 we have added a figure with a schematic showing the elements discussed in it (Fig. 2,3,4 and 11). The new figures are designed to support the reading experience of the article by summarizing the information presented in each section, and to provide a graphical navigation map of the ecosystem. We believe these additions significantly improve the readability and usability of the paper and move us closer to the original goal of providing an accessible overview of the ocean data ecosystem. We thank the reviewer for this excellent suggestion, which has greatly strengthened the paper.**
- **For the purpose of providing an interactive map which may be kept up-to-date for a long-term reference, we created an online ocean data ecosystem ontology (*https://odini.net/OceanDataEcoSystem/, with an interactive visual representation available at https://webvowl.odini.net*). The on-line tool is open access, and may be updated and extended, thus remaining relevant over time. The additional tool is referred in the text, as follows (lines 184-188): "*For the purpose of providing an interactive map, we created an online ocean data ecosystem ontology available at https://odini.net/OceanDataEcoSystem/, with an interactive visual representation available at https://webvowl.odini.net. This map is a long-term reference that may be updated and extended. To facilitate contributions and comments from the public, we make the ontology publicly available on GitLab (https://gitlab.com/odini_dev/data-ecosystem-ontology) and invite readers to suggest additions and corrections (ODINI / Data Ecosystem Ontology · GitLab, n.d.; Ontology Documentation generated by WIDOCO, n.d.; WebVOWL, n.d.).*".**

Section 3 of the paper elaborates on the different elements of the Ocean Data Ecosystem model, however this section is 25 pages long, stepping to sub-sub-sub sections. This quickly become hard to read, follow and the main message of the paper is lost. Summarising this information into shorter sections, like Section 3.1 Stakeholders, and providing the additional information through a table, appendix or supplementary material could be a better way to convey this information without losing the goal of the manuscript. This section also follows a different order to the model provided in Figure 2, Stakeholders (Section 3.1), Societal elements (Section 3.2), Data sources and product offerings (Section 3.4), Standards and best practices (Section 3.3), and Emerging technologies (Section 3.5). Different terminologies are also used, e.g., e.g., emerging technologies vs emerging solutions).

**Answer: While acknowledging that section 3 is long, we think it contains important information that will be very useful for the ocean science community. As detailed above, to improve the clarity and readability of the paper as a whole, and of this section particularly, we have added figures (Fig. 2,3,4 and 11) that summarise the content of the different subsections in an easy-to-navigate manner. Since the reader can now follow the navigation map and decide where to focus the reading, we left the textual descriptions of the different elements as appear in the different subsections untouched.**

**We checked the text for consistency. Specifically, the term "emerging technologies" was changed to "emerging solutions" throughout the text.**

**The new version of Fig. 1 now refers to the different elements of the ocean data ecosystem in the same order as they appear in the manuscript.**

A very long list of data sources, product offerings, interoperability tools and frameworks, and emerging solutions is provided but this list is not exhaustive nor is it clear why some has been chosen to be included but not others. This section would be significantly improved by providing examples to explain the element in the model rather than presenting an exhaustive list which will inevitably miss things and quickly be outdated. However, an extensive list is a useful resource, so perhaps this again could be provided as a digital resource to accompany this manuscript which could be updated beyond the publication of this manuscript to remain in date.

**Answer: We thank the reviewer for this insightful comment. A number of revisions were made to address the drawbacks that were pointed in it. First of all, we followed the reviewer's suggestion and implemented a new web-based tool accompanying the paper, with an interactive visual representation available at https://webvowl.odini.net. This map provides the reader a long-term reference, open, and may be updated and extended. The web-based tool will be updated regularly, maintaining the paper up-to-date after its publication. This complementary element of the paper and its implications are described in the following texts, which were added to the manuscript:**

**Lines 183-184: "*For the purpose of providing an interactive map, we created an online ocean data ecosystem ontology available at https://odini.net/OceanDataEcoSystem/, with an interactive visual representation that is available at https://webvowl.odini.net.***

*This map is a long-term reference that may be updated and extended. To facilitate contributions and comments from the public, we make the ontology publicly available on GitLab (https://gitlab.com/odini_dev/data-ecosystem-ontology) and invite readers to suggest additions and corrections (ODINI / Data Ecosystem Ontology · GitLab, n.d.; Ontology Documentation generated by WIDOCO, n.d.; WebVOWL, n.d.).*"

**Lines 1072-1076:** "*To maintain long-term relevance, the ecosystem model presented is aimed to be used as a tool in further characterization of the ocean data ecosystem. The examples given are not exhaustive, and the reader may further identify relevant examples within their domain. The model has been placed as an open online resource as an ontology, describing the elements of the data ecosystem as classes, and examples as instances with relationships. The model and ontology are open to be further validated and refined by the ocean data community*"

**We agree with the reviewer that description of data sources, tools, and frameworks is not exhaustive. Due to the very broad nature of the ocean data ecosystem, we don't think it is feasible to cover the large number of elements it contains, and the elements that are presented are given as representative examples. A clarification of this point and an improved of the approach we take, are found in these paragraphs:**

**Lines 176-182:** "*Rather than presenting an exhaustive and potentially quickly outdated inventory, the model is intended to help readers identify and characterize relevant examples (e.g. stakeholders, societal elements, integration tools, data sources and emerging solutions) that are most applicable to their specific domain, use case, or geographic context.  By presenting a flexible and structured framework, the model will serve as a tool to enable to add new developments and technologies, as they emerge in the ocean data ecosystem.*"

**Lines 189-193:** "*The methodology we used is based on a thorough literature and website review of over 90 scientific articles and over 100 websites. Articles and examples selected to illustrate the elements of the model have been selected based on using search terms such as 'ocean data,' 'ocean data interoperability,' and 'marine ontologies.' The examples are not exhaustive and are intended as a starting point for ocean data professionals, who are encouraged to explore additional resources specific to their research areas.*"

This list and the entire paper seem to largely ignore regional efforts in ocean observing. For example,  IOOS is often referenced but not other GOOS Regional Alliances. AtlantOS is referenced in section 3.4.10 but it is unclear why this has been pulled out but not other regional observing systems and their products such as the Southern Ocean Observing System (SOOS) data product, SOOSmap (soosmap.aq).

**Answer: As discussed above, and clarified in the revised version, this paper is not meant to provide an exhaustive list of the ocean data ecosystem elements, but rather to provide a model that is accompanied by representative examples. Accordingly, we provide a number of representative examples for regional efforts, including the EOOS (European Ocean Observing System Framework), EuroBIS (European Node of the**

**international Ocean Biodiversity Information System, and AtlantOS ("AtlantOS - EuroGOOS).**

It is not clear what makes all those listed in Section 3.5 emerging technologies. Some of those repositories and portals mentioned in 3.4 can very easily be included here as well, given the massive push globally to be more interoperable and implement these emerging technologies across all platforms. The "emerging technologies" section would better focused on the strategies and approaches and provide some examples of programs/institutions that are applying those to their already existing (or new) platforms.

**Answer: As detailed above, the term emerging technologies was changed to "emerging solutions". We agree there is no clear separation line between data sources and emerging solutions, such that some of the data sources may very well provide innovative new approaches for managing ocean data interoperability, and the emerging solutions may be considered data sources of the ocean data ecosystem.**

**That being said, the emerging solutions were selected to include examples of different approaches, not necessarily the leading and most prominent efforts, but interesting use cases that approach the problem of data interoperability of the ocean data ecosystem by implementing different approaches such as cloud, developing a new ocean data platform (e.g. Hub Ocean), ai solutions for data interoperability (e.g. ODINI) etc.**

Section 3.4 Data Sources and Product Offering, is difficult to follow given the order in which things are presented and the language of categories not being consistent. Section 3.4 outlines three categories of data sources (raw source, repository or portal) but the following sub-section doesn't seem to follow these. E.g., Section 3.4.1 WOD, the data source category chosen is not one detailed in the paragraph above. Consistent language needs to be used. Same goes for Section 3.4.6 CMEMS. Section 3.4 is also in a different order to what is presented in Figure 2.

**Answer: The data sources described in section 3.4 are organized according to the three infrastructural approaches distinguishing them, namely centralized, distributed, federated. For each approach we provide several representative examples. This rationale is now explicitly represented in Fig. 4, which summarizes the ocean data sources described in section, emphasizing their are organization according to their infrastructural approach. To further improve the clarity of this section, the following text was added (lines 486 - 487): "Here we categorize the different data sources by their infrastructural approach, as defined in section 2.1, namely: centralized, federated, and distributed data ecosystems. We now give an overview on some of the major data sources (Fig. 4)."**

Specific Comments:

Figure 1. The description provided in the text is great but the figure does not convey anything and seems redundant. I suggest removing this figure from the manuscript.

**Answer: We agree with the reviewer and have removed the figure from the revised version of the manuscript.**

Terminologies and language varies throughout the manuscript. Consistent use of terminologies and language is highly recommended.

**Answer: we have done a thorough reading of the manuscript, correcting inconsistencies and terminology differences.**

Section 4 starts with a sentence saying there are 3 main concepts for a data ecosystem, yet the beginning of the paper clearly outlines 4 main concepts. This section should also move earlier in the paper such as before section 3 as it provides the overview components of the data ecosystem being presented in this manuscript.

**Answer: The notion of 3 main concepts is erroneous, and was corrected to 4 main concepts. While we understand the rationale of placing the section on roles in the ecosystem (section 4), prior to the section on Modelling and mapping the ocean data ecosystem (section 3), we think the order fits better, as section 4 cites information presented in section 3.**

Lines 102-103. Having specific examples of organisations that adhere to each centralized, federated or distributed data ecosystems might be helpful for readers to understand this. Or use words other than those in the architecture titles to describe them (e.g., not using centrally to describe a centralised ecosystem).

**Answer: as noted above, the data source examples are now aligned according to these categories and are further exemplified in section 3. Furthermore, we refined the definition In line 104, and removed the redundant definition in section 5.**

Line 124-132: This sentence is very long, complex and hard to follow. Please break into multiple sentences to make it clearer to the reader.

**Answer: The sentence has been rephrased as follows: "*A comprehensive overview of the ocean data sector has been provided by Tanhua et al., (2019a, 2019b, 2021), who reviewed recent developments in the technical capacity and requirement setting for a data management system in the frame of the Global Ocean Observing System (GOOS). These papers emphasize the importance of well-managed data management systems for ensuring the data collected by the ocean observing systems are accessible for current and future uses.*".**

Technical Corrections:

An overall copy-edit of the paper is needed to improve grammar, check for missing or additional spaces before/after brackets, inconsistent capitalisation etc.

**Answer: grammar and editing errors were corrected throughout the text.**

"The concept of a data ecosystem in ocean research" header on line 114 needs to be numbered.

**Answer: The following headers were added to section 2 and numbered:**

> **2.1 Data ecosystem general definitions and examples (line 75)**
> **2.2   The concept of a data ecosystem in ocean research (line 111)**

Figures 2 & 3. Any acronyms provided in the figures need to be described in the figure heading.

**Answer: where appropriate acronym description was added to the figure captions**

Section 3.2.2 Key Initiatives needs to have its own header. The header cannot be part of the first sentence of the paragraph. Same goes for section 3.2.3, Section 3.2.4, and Section 3.4.10.

**Answer: headers were added accordingly.**

Throughout there are bolded headers without numbers, this may be due to the journal requirements but this make it difficult to read. These either need to be numbered and have appropriate headers keeping the formatting of the rest of the paper, or have the bolded format of the text be removed. Lines 427, 435, 447, 461, 490, 510, 527, 612, 637, 648, 686, 714, 746, 823, 828, 834, 855, 862, 868, 886, 900, 903.

**Answer: bold headers have been removed throughout the text**.

---

## Referee Report (RR1)

The authors have addressed all by feedback and I thank them for the constructive response to the review process. Also, I find the new Kumu visualisation of the data ecosystem compelling. Thank you again.

---

## Author Response (AR2)

We thank the reviewers for their constructive comments and suggestions. We have thoroughly addressed all the points raised and implemented the suggested change. Importantly, following the insightful suggestions made by the reviewers we have:

- Clarified throughout the text that the ecosystem elements on which we chose to elaborate are merely representative examples.
- Changed the approach for interactive representation of the ocean data ecosystem, using a comprehensive knowledge graph instead of an ontology.
- Added information that was lacking in the previous version of the manuscript.
- Elaborated our vision for the future of the ocean data ecosystem.

Our detailed point-by-point response to the reviewers' comments and suggestions is found below. All page/line/reference/figure numbers refer to the clean version of the revised manuscript. Reviewers' comments are in regular text, **our responses are in bold** and ***new text from the manuscript is in bold italics***.

**Reviewer #1**

This manuscript is much improved with a slightly clearer structure and the addition on online tool is fantastic for a resource that can be kept up-to-date to ensure relevance of this manuscript and tool into the future. However, my fundamental concerns regarding the length and breadth of the paper remain.

The large and detailed list of "examples" in this manuscript mean the overall aim of the publication is lost and the map of the data ecosystem isn't clear. Whilst the authors have made it clearer these are "examples", the detail on these examples makes it appear more like an exhaustive list, which it isn't. Further, these examples are included in the online tool but no where does it make this clear these are "examples only" therefore again it appears to be an exhaustive list, which it isn't.

**Answer: We thank the reviewer for the thorough reading and the constructive comments. We agree that there is somewhat a discrepancy between the length of the paper, and the fact that it does not give an exhaustive description of the ocean data ecosystem and all its components. We do think however that expanding the discussion on a limited number of selected examples, while clarifying the fact that they are such, does provide valuable information that will help the reader navigate his way in the world of ocean data, thus facilitating data-driven ocean research. Below we explicitly address the reviewers suggestions and comments.**

My suggestions to make this manuscript clearer and more impactful:

1) Ensure it is very obvious that the description of data sources, tools and frameworks are examples only and not an exhaustive list.

**Answer: we agree with the reviewer that it is essential to clarify that the different ecosystem elements discussed in detail are only, and do not constitute an exhaustive**

list. Accordingly, in the current version of the manuscript this is mentioned explicitly a number of times:

Line 182: "Rather than presenting an exhaustive and potentially quickly outdated inventory…".

Line 190: "The methodology we used is based on a thorough literature and website review of over 90 scientific articles and over 100 websites. Articles and examples selected to illustrate the elements…".

Line 1138: "To maintain long-term relevance, the ecosystem model presented is aimed to be used as a tool in further characterization of the ocean data ecosystem. The examples given are not exhaustive…".

To further clarify this point, in the revised manuscript we have added the following text to section 3.4. (lines 510-514):
"The description of data sources, tools, and frameworks is not exhaustive. Due to the very broad nature of the ocean data ecosystem, rather than covering the large number of elements it contains, we give several representative examples. The readers may use these examples, presented in detail, to broaden their knowledge of the data ecosystem, and may perform further review of other data sources within their respective fields. We used a framework for analyzing the data sources, including key characteristics composing a data source, namely: Organization details such as number of partnering organizations, oceanographic domain, geographic region, number of data sets, data catalogue and product offering, main uses by the specific user community and interoperability strategy. Other characteristics may be selected depending on the reader's focus and interests."

2) Ensure all figures state that the list are examples (e.g., Figure 4 caption to include wording similar to Figure 11).

**Answer**: we have made sure that the relevant figure captions clearly state the exemplary nature of the figures, as follows:

"Figure 2. A diagram summarizing the types of societal elements and the representative examples discussed in Sec. 3.2"

"Figure 3. A diagram summarizing the types of interoperability tools and the representative examples discussed in Sec. 3.3."

"Figure 4. A diagram summarizing the examples for ocean data sources discussed in section 3.4. The exemplified data sources are organized according to their infrastructural approach (namely federated, distributed and centralized)."

"Figure 11. A diagram summarizing the examples for emerging solutions discussed in Sec. 3.5, along with brief descriptions of the approach they represent."

3) Significantly strip back example sections e.g., 3.4.1-3.4.10 to be summary paragraph or two in section 3.4 and instead include the detail in a supplementary section if the authors wish to still make this available

**Answer: Despite its length, we do not think that stripping back the example sections will improve the paper. The selected examples are sufficiently representative to enable readers to develop a product-oriented perspective, and provide valuable information for readers who wish to understand and characterize key products in this domain. While a different selection of examples may have been made, our evaluation suggests that similar conclusions would emerge. Importantly, the examples reveal a connecting thread across the different systems: how information is organized into product catalogs, how dataset searches are supported, how information remains distributed across disciplines and platforms, and what gaps still persist. Moreover, the organization of the selected examples in consequent subsections allows readers who are more familiar with the domain to skip their description without influencing the paper's consistency and readability. We therefore believe that retaining the examples in the main text adds significant value.**

I also think some of the explanatory text in the response to reviewers could be useful to include in the manuscript e.g., emerging solutions explanatory text.

**Answer: we have added explanatory text as suggested in section 3.4. (see details in our response above).**

As the manuscript stands, whilst it is factually correct and could be published as-is, I would worry on the impact the paper will have give its length and breadth. A stripping back of the paper and focus just on the key message of the easy to navigate map of the ocean data ecosystem would greatly improve the usability and impact of this tool, which is a very useful contribution to the ocean data community

**Answer: We thank the reviewer for the thoughtful comment regarding the length and breadth of the manuscript. We acknowledge that the paper is relatively long. However, it has been deliberately organized into clearly delineated sections and subsections so that readers can readily focus on areas of specific interest or skip sections that are less relevant to them while still following the overall narrative. We believe this structure preserves readability and flow, while allowing the manuscript to serve both readers seeking the key message (i.e. a map of the ocean data ecosystem) and those wishing to explore additional context and details.**

Thank you for producing a revised version. The new version addresses some of my feedback on specific points raised it also introduces or highlights new issues to be addressed before I can recommend the manuscript for publication.

**Answer: We thank the reviewer for the thorough evaluation process, and appreciate the thoughtful and knowledgeable comments. As detailed below we have carefully addressed the reviewer's concern, thus substantially improving the quality of the paper.**

On second review and after revisions the context behind and underpinning many of the entities described in the paper seems ambiguous or confused. Three examples that stand out are:

1. Of the principles presented (FAIR, TRUST, CARE); FAIR is given precedence with TRUST and CARE also existing. The reality is that each is designed to serve a different purpose and applicable in different contexts e.g. FAIR is about data accessibility and utility, TRUST is about the sustainability and practices used in data curation with the connection between CoreTrustSeal being an accreditation for TRUST missing later in the manuscript, and CARE is critical when handling data when indigenous communities are an interest-holder. Thus, each has a place and a role and value in its own right in their contexts.

**Answer: we thank the reviewer for drawing our attention to this important issue. To give a more representative picture of the three principles and their roles, the relevant paragraph was modified as follows (lines 232-247):**

*"By and large, ocean data management is guided by the FAIR, TRUST and CARE principles, each is designed to serve a different purpose and applicable in different contexts. The FAIR principles are mainly concerned with scientific data management and stewardship and are meant to ensure reusability of data, with an emphasis on enabling the automation of data findability and usability (Tanhua et al., 2019b; "The FAIR Data Principles – FORCE11," n.d.; Wilkinson et al., 2016). The TRUST principles of data repositories (Lin et al., 2020), deals with data curation, providing guidance to demonstrate transparency, responsibility, user focus, sustainability, and technology. The CARE principles ("CARE Principles — Global Indigenous Data Alliance," n.d.) were defined for ensuring proper handling of data associated with indigenous communities, defining measures for Collective Benefit, Authority to Control, Responsibility, and Ethics.*

*In our context of modeling the ocean data ecosystem with a focus on data interoperability, we find the FAIR principles a useful tool, as they are meant to provide*

*data producers and publishers measurable guidelines for ensuring their data implementation to be Findable, Accessible, Interoperable, and Reusable, to overcome the barriers for large scale data utilization. The need to follow the FAIR principles stems from the fact that gathering the data required for answering research questions is often a tedious and time-consuming task, largely due to lack of attention to how the data assets are preserved when they are created (Tanhua et al., 2019b). The FAIR principles answer to the step-by-step process by which machines will be able to process the data, identifying the relevant data within a given context, determining if it is useful, if it is usable in terms of license or other accessibility and taking action (Tanhua et al., 2019b)."*

In addition, we clarify in the introduction that our ocean data ecosystem model was developed in the context of our focus on data interoperability (lines 66-70):

*"To this end, this paper presents an overview of the ocean-data ecosystem and proposes a model that maps its key actors and their roles, with particular attention to data platforms and interoperability. The discussion is framed from the perspective of the marine researcher as end user, with the overarching goal of facilitating synergetic work between the actors involved in different aspects of ocean data, thus improving the ability to utilize the vast amount of available ocean data to address important scientific questions".*

2. A key subtlety lost in the manuscript is that each data entity or service described has a different user community with duplication of data widespread across the ecosystem. For example, in this context many of the data assets in WOD are an aggregation of subsets of data from SDN/PANGAEA/NOAA/etc but they have been curated into a product for a specific purpose and user community where WOD is the modern equivalent of a hydrographic atlas. This duplication of assets is ubiquitous in the ecosystem e.g. EMODNet aggregates data form many sources (SDN, ICES, CMEMS, etc) to build products for specific applications and users.

3. The role of DataONE and SeaDataNet overlap significantly with one being a US entity bringing together American data assets and the later being a European entity. Albeit they have different funding, governance and collaboration agreements but the similarity in their high-level functions for end users is not clear in the revised manuscript.

**Answer: We thank the reviewer for drawing our attention to these important points. To address the missing information identified in these two comments, we added the following text to our general description of the data sources and referenced the relevant sections where these sources are discussed (lines 492-507):**

*"Notably, different data sources may exhibit significant similarity, often with overlap between their contents. In general, each data source serves a distinct user community, and datasets may be duplicated throughout the ecosystem. For example, many records in the World Ocean Database (WOD, see below Sec. 3.4.1) are compiled from subsets of data originating in datasets such as PANGAEA (see below Sec. 3.4.5.), but are curated into a product designed for a specific purpose and audience. Overlap and similarity between data sources can be exemplified for the cases of SeaDataNet (see below Sec. 3.4.3) and DataONE (see below Sec. 3.4.7), which although differing in their funding structures, partnership frameworks and infrastructural approach, align in their high-level functions for end users."*

In my initial review I raised the lack of vision presented in the manuscript for the next 20 years based on the special issue scope, This feedback does not appear to be addressed significantly by the authors with the manuscript presenting current trends in the summary section.

Answer: **We have elaborated our vision for the future of the ocean data ecosystem by adding the following text to the summary and conclusion section (lines 1107-1137):**

*"Looking ahead over the next two decades the ocean data ecosystem is set to undergo further transformations, driven primarily by dramatic growth in the amount and diversity of oceanic data, and by rapid technological developments. The expected increase in data availability and diversity is a natural continuation of the growing use of autonomous and remote sensing platforms, expansion of global observation networks, and improved ability to collect and analyze new data types suchas environmental DNA and underwater imagery. Advances in data collection methods results in an unprecedented influx of ocean data each day, often in real-time, propelling ocean research into the era of big data, characterized by vast volumes, diverse formats, and widely dispersed datasets (Tanhua et al., 2019).*

*The synthesis presented in this review points to a decisive transition: the ocean data ecosystem is evolving from a patchwork of independent repositories into a globally connected, service-oriented network. Interoperable standards, shared vocabularies, and federated cloud infrastructures are dismantling the old "portal–download" paradigm and enabling machine-actionable data flows across disciplines and borders. The next decade will likely see near-real-time discovery, access, and fusion of ocean observations - from autonomous sensors to satellite archives - within a seamless digital environment.A defining feature of this emerging landscape is the integration of advanced artificial intelligence (AI) and machine-learning methods. As the volume, velocity, and variety of ocean data continue to grow, AI is becoming indispensable for automated quality control, feature extraction, and pattern recognition across heterogeneous data streams. Deep-learning models can already detect mesoscale*

*eddies, track marine heatwaves, and identify biodiversity "hot spots" in vast image libraries. Looking ahead, AI-driven digital twins of the ocean will couple observation networks with predictive models to deliver near-instant forecasting and scenario testing, transforming both basic research and operational decision-making.*

*This technological leap will also reshape the social and governance dimensions of the ecosystem. Trust, transparency, and inclusivity, embodied in the FAIR, CARE, and TRUST principles, remain critical as AI systems begin to make or recommend management choices. International initiatives such as the UN Ocean Decade, the Ocean Data Action Coalition, and IOC/IODE programs are fostering open access and shared stewardship, while highlighting the need for clear policies on data provenance, algorithmic accountability, and equitable participation. Ethical AI frameworks, together with persistent identifiers and accreditation schemes, will help maintain confidence in automated analyses and encourage collaboration across nations and institutions.*

*Ultimately, the ocean data ecosystem is poised to become an active, intelligent engine for discovery and policy. By linking high-resolution observations, interoperable standards, and AI-powered analytics, it will enable rapid synthesis of knowledge for climate adaptation, biodiversity conservation, sustainable fisheries, and the broader blue economy. In this envisioned future, ocean data are not merely archived, they are continuously analyzed, interpreted, and applied, allowing scientists, governments, and society to anticipate and respond to a changing ocean with unprecedented speed and precision."*

The revised version introduces a significant issue into the manuscript. The revised version attempts to build on the previous version by describing the eco-system as an ontology. I have a good awareness of ontologies but am not an expert on the subject so sought advice from three experts in different contexts; specifically global ocean data brokering, and industry. I sought feedback based on the openly available published ontology excluding the manuscript to preserve manuscript confidentiality. A summary of the feedback from these specialists is below. This feedback is critical to address for the work to be of publication quality. From my perspective as a reviewer the value and interesting element of the original submission was in its broad context rather than the ecosystem model. The ecosystem model provided the structured context as a wrapper for the rest of the manuscript making a complex system understandable to the reader. Based on the expert feedback publishing the model as ontology in its current form adds little value unless it builds existing domain knowledge and is technically sound in its formulation. Additionally, A mature relational graph between entities has the potential to address points 2) and 3) raised above.

I also disagree with the final conclusion in the revised in manuscript. The original manuscript was an analysis of the existing data ecosystem using the relationship model to help the

reader readily navigate and understand what is an inherently a complex collection of data nodes and brokered data because of numerous dependencies including geopolitical requirements, requirements of varied user communities and collaborations, different funding environments, and human behaviour. The revised manuscript presents the pre-existing ecosystem as a new concept which is a questionable conclusion with the ecosystems having to exist already for this review paper to be possible (with the citied literature demonstrating pre-existence of an ecosystem). To me as the reviewer novelty and value in the first version was in helping a researcher navigate the complex data ecosystem that exists in the ocean community which has been somewhat lost in the revised manuscript.

Ontological feedback based solely on the openly published ontology at .

Global ocean data brokering context

Foremost, this isn't really an ontology: the class-subclass relationship is not ontological, in that subclasses do not inherit all properties of superclasses. In fact, they're often completely different things.

For example, a stakeholder can't be an OWL subclass of an "ocean digital ecosystem". That's like saying a chair is a type of house. An industry stakeholder can be a type of stakeholder, so that's right. It's a little patchy, and if this was submitted to me by a student in an ontology class, I wouldn't pass them.

Another thing that's missing is any consistent and logically coherent definitions. So most of the semantics here are roughly implied. This is more a mind map.

• Are there strong or excessive overlaps with existing ontologies?

As this is not an ontology, I don't think this is the right question. However, there are strong overlaps with resources like ODISCat and Ocean Expert which already map many components of the ocean digital ecosystem. This is especially true as they now release their content in JSON-LD using schema.org semantics, which gives them more semantic rigour than the resource you're reviewing.

OE and ODISCat are definitely not a perfect resources, so I would see merit in a research group grabbing its JSON-LD/RDF and creating a curated / improved product.

One could also point to regional maps of organisations and stakeholders, like EDMO and EDMERP, even MEDIN's offerings as an ODIS Node or independently.

• Is there alignment with concepts in similar or related ontologies needed in this ontology?

They should not attempt an ontology construction here - I don't think it will help them. It seems that they would make more progress with building a relaxed knowledge graph as an example of how to map a digital ecosystem.

However, I note again, much of this is redundant with ODISCat and Ocean Expert, especially when combined with ORCID, ROR, and other systems. It would be distracting and counterproductive for another group to claim they are running an authoritative map of the ecosystem.

More broadly, if the paper claims to be about some "new" concept of digital / data ecosystems, please do remind them this is not new and many actors have been working on it.

Industry context

What you've shared is not an ontology, it just happens to use an ontology modelling language to do something hopefully useful for them. Then there are the fundamental issues in the misuse of the relations; they have used subclass more like a partOf relation and then failed to produce any descriptions of the classes they have defined. From a FAIR perspective it fails on a number of fronts.

There are some quite major issues in understanding what they expect to get from this work as well. I would suggest looking at something like Kumu if they have not already done so to deliver research / community value, however it sounds like we would also have concerns in this also from a perspective of being novel vs other efforts. Other products are available, but it's the one I've seen The Alan Turing Institute use successfully. They should look towards a concept map to understand who is doing what and using what in the landscape with questions in mind of what they would like to answer rather than graphs for technology and visual interest.

**Answer: We thank the reviewer for taking the pains to obtain additional expert reviews from Ontology experts and apologize for the unexpected time and effort this incurred. The incentive for development of the ontology came from the suggestion of reviewer #1, and was meant to provide an online tool that can be kept up-to-date to ensure relevance of the manuscript. Thanks to this thoughtful comment, we realize that it was presumptuous of us to elevate our model to an ontology without rigorous analysis of existing ontologies and a more accurate integration of the suggested models within the existing ontology landscape. While we believe this should be our eventual goal, we agree with the reviewer that the current state is premature and inaccurate. We defer the creation of a comprehensive knowledge graph, based on the ontologies suggested by the reviewer and, if necessary, a minimal extension of these, to future work.  We thank the reviewer for the suggestion to use a more loosely defined relation model and the Kumu tool for visualizing the model. We have done so and provided a description, and a reference to a public Kumu model instead of the Ontology in the previous revision. Accordingly, we have revised the manuscript as follows (lines 187-195):**

"*For the purpose of providing an interactive map, we created an online ocean data ecosystem relational model available at [https://kumu.io/odini/ocean-data-ecosystem](https://kumu.io/odini/ocean-data-ecosystem) the map is a long-term reference, open, and may be updated and extended. To facilitate contributions and comments from the public, we make a public issue system available [https://gitlab.com/odini_dev/data-ecosystem-ontology](https://gitlab.com/odini_dev/data-ecosystem-ontology)) and invite readers to suggest additions and corrections .*

*The methodology we used is based on a thorough literature and website review of over 90 scientific articles and over 100 websites. Articles and examples selected to illustrate the elements of the model have been selected based on using search terms such as 'ocean data,' 'ocean data interoperability,' and 'marine ontologies.' The examples are not exhaustive and are intended as a starting point for ocean data professionals, who are encouraged to explore additional resources specific to their research areas.*"

In addition, the concluding paragraph of the paper was modified as follows (lines 1137-1149):

*"In summary, by mapping the market landscape in the field of ocean data, this review paper is meant to enable the reader, especially the new entrant to the ocean data field, to establish an understanding of the ocean data sector. To maintain long-term relevance, the ecosystem model presented is aimed to be used as a tool in further characterization of the ocean data ecosystem. The examples given are not exhaustive, and the reader may further identify relevant examples within their domain. The model has been placed as an open online resource, describing the elements of the data ecosystem as concepts, and examples as instances with relationships. The model is open to be further validated and refined by the ocean data community. The results bridge gaps between different disciplines and levels of familiarity with ocean data. We provide an up-to-date analysis of ocean data sources and emerging solutions and a summary of relevant data standardization efforts such as marine standards, vocabularies, and ontologies. By characterizing the ocean data ecosystem, we intend to assist the scientific community in identifying the gaps, current needs and future vision of the ocean data ecosystem. This work aims to contribute to the development of needs-based solutions, components, products, services, and technologies, thus contributing to the evolution of the ocean data ecosystem and promoting data-based ocean research."*